# Smooth And Consistent Probabilistic Regression Trees

**Sami Alkhoury**
Univ. Grenoble Alpes, CNRS, Grenoble INP, LIG, Grenoble, France
`sami.alkhoury@univ-grenoble-alpes.fr`

**Emilie Devijver**
Univ. Grenoble Alpes, CNRS, Grenoble INP, LIG, Grenoble, France
`emilie.devijver@univ-grenoble-alpes.fr`

**Marianne Clausel**
Université de Lorraine, CNRS, IECL, Nancy, France
`marianne.clausel@univ-lorraine.fr`

**Myriam Tami**
Univ. Paris-Saclay, CentraleSupélec, MICS, Gif-sur-Yvette, France
`myriam.tami@centralesupelec.fr`

**Eric Gaussier**
Univ. Grenoble Alpes, CNRS, Grenoble INP, LIG, Grenoble, France
`eric.gaussier@imag.fr`

**Georges Oppenheim**
Univ. Paris-Est-Marne la Vallée, Département de Mathématiques, Marne-la-Vallée, France
`georges.oppenheim@gmail.com`

## Abstract

We propose here a generalization of regression trees, referred to as *Probabilistic Regression (PR)* trees, that adapt to the smoothness of the prediction function relating input and output variables while preserving the interpretability of the prediction and being robust to noise. In PR trees, an observation is associated to all regions of a tree through a probability distribution that reflects how far the observation is to a region. We show that such trees are consistent, meaning that their error tends to 0 when the sample size tends to infinity, a property that has not been established for similar, previous proposals as *Soft* trees and *Smooth Transition Regression* trees. We further explain how PR trees can be used in different ensemble methods, namely Random Forests and Gradient Boosted Trees. Lastly, we assess their performance through extensive experiments that illustrate their benefits in terms of performance, interpretability and robustness to noise.

## 1 Introduction

Classification and regression trees (CART) [3] and the ensemble methods based on them, as Random Forests [2] and Gradient Boosted Trees [11, 8], have been successfully used for regression problems

in many applications and machine learning competitions. Indeed, in a 2017 survey conducted by Kaggle[1], decision trees and random forests are respectively the second and third most used machine learning methods in industries after logistic regression. It is however well known (see, *e.g.*, Irsoy et al. [14], Linero and Yang [20]) that standard decision/regression trees, based on piece-wise constant functions with hard assignments of data points to regions, may have difficulties to adapt to the smoothness of the link functions as well as to the noise in the input data.

We address these problems in this study by adapting standard regression trees through smooth predictions based on probability functions that relate each data point to each region of the tree. The trees thus obtained can naturally adapt to noisy input and can be shown to be consistent, a property that was not, to the best of our knowledge, established for previous, similar attempts. In addition, we show how to use such trees in ensemble methods as Random Forests and Gradient Boosted Trees. Our contributions are thus fourfold: (1) we introduce new regression trees, called *PR* trees for *Probabilistic Regression* trees, that can adapt to noisy dataset as well as to the smoothness of the prediction function relating input and output variables while preserving the interpretability of the prediction and being robust to noise; (2) we prove the consistency of the PR trees thus obtained, (3) we extend these trees to Random Forests and Gradient Boosted Trees and (4) we show, experimentally, their benefits in terms of performance, interpretability and robustness to noise. There is however *no free lunch*, and these additional properties come with a computational cost, as described in Appendix A.2.

The remainder of the paper is organized as follows: Section 2 discusses the related work. Section 3 then introduces PR trees and their inference. Section 3 presents the main results concerning their consistency, the complete proof of which can be found in the Supplementary Material. Section 4 presents the experiments conducted on PR trees and discusses their extension to Random Forests and Gradient Boosted Trees. Finally, Section 5 concludes the paper.

## 2   Related Work

Several researchers have tried to adapt decision trees so as to explicitly take into account the noise and uncertainty present in the input data. For example, Fuzzy Decision Trees [4, 30, 15, 22], designed for classification purposes, assume that the values for some features and classes are associated with membership functions that allow one to associate an example to different rules and predict several classes (with various degrees) with a rule. The approach presented in Ma et al. [21] fits within the same framework and aims at reducing data uncertainty by querying adequate data while learning the tree. The possibilistic trees considered in Elouedi et al. [9], Jenhani et al. [16] also follow the same principles, the uncertainty being this time modeled with belief functions. Another approach to model uncertainty in the input is based on Uncertain Decision Trees [28, 23, 19], also designed for classification, that assume that examples can take on several values through explicit probability density functions (pdf) on given intervals (or explicit probability tables for categorical values). An example may belong, with a certain probability, to several leaves depending on its value range. All these approaches rely on assumptions on the information available for each input (membership scores and pdfs or probability tables) and do not directly aim at adapting to the smoothness of the true prediction function.

Closer to our objectives are Soft trees, introduced in Irsoy et al. [14], Fuzzy Trees, introduced in Suarez and Lutsko [26], and Smooth Transition Regression trees (STR trees), introduced in [6]. Soft and Fuzzy trees are very close to each other: they can both be used for classification and regression, and learn a parameter vector at each node, the dimensionality of which equals the one of the input data. This vector is used in a gating function, based on the sigmoid function in both cases, assigning, to each example, a probability to branch to the left and right children of the node. Each example is thus assigned to all leaves with a certain classmembership, and the final prediction is a smooth combination of the prediction at each node. Soft and Fuzzy trees can be seen as a direct extension of hierarchical mixtures of experts (HMEs, Jordan and Jacobs [17]): indeed, if HMEs either use predefined trees or trees learned by another method (typically a standard decision/regression tree), soft trees are constructed while learning the hierarchy of experts. STR trees follow the same general principle but rely on a single parameter at each node. A sigmoid-based gating function is also used to assign points to different regions of the tree. Adaptive Neural Trees [27] and Deep Neural Decision

Forests [18] are both built from decision trees. These models are very close to soft trees; however, the models are enhanced with a neural network representation and suffer from a lack of interpretability (one can even argue that these models are not tree models *per se*). The main differences between these approaches and PR trees lie in the way the assignment of examples to leaves is done, in the number of parameters used by each method and in their theoretical guarantees. By directly relying on a probability distribution, PR trees rely on less parameters which make them *a priori* more robust to overfitting. In addition, we show here that PR trees are consistent; we know of no such results for the other models. Frosst and Hinton [12] consider a specific variant of the soft tree model, with knowledge distillation, which is beyond the scope of this work.

Lastly, one of the objectives of this study is to use PR trees in ensemble methods. The most well-known ensemble methods built on top of regression trees are certainly Random Forests (RF)[2] and Gradient Boosted Trees (GBT) [11, 8]. We show in Section 4 how to use PR trees in these ensemble methods and illustrate their behavior compared to standard RF, GBT and BooST [10], a boosting-based ensemble extension of STR trees. More recently, tree-based Bayesian ensemble methods, as BART [5] and soft-BART [20], have been proposed. If these methods are interesting, they fit within a (more complex) Bayesian framework that needs to be adapted to the different trees used (this contrasts with Random Forests and Gradient Boosted Trees which can readily integrate different regression trees). Their consideration is beyond the scope of the current study.

## 3 Probabilistic Regression Trees

Let $\mathbf{X} = (X_1, \cdots, X_p)$ be a $p$-dimensional input random vector lying almost surely in a compact subspace $\mathcal{X}$ of $\mathbb{R}^p$, and let $Y$ be an output random variable linked to $\mathbf{X}$ through:

$$Y = f(\mathbf{X}; \Theta) + \varepsilon_Y, \ \varepsilon_Y \sim \mathcal{N}(0, \tilde{\sigma}^2), \tag{1}$$

where $\Theta$ is the set of parameters on which $f$ relies.

Due to their success, we are interested in this study in approximations of $f$ based on regression trees, either considered in isolation or aggregated in *sum-of-trees* models as Random Forests or Gradient Boosted Trees. Regression trees [3] define a partition of $\mathbb{R}^p$ into $K$ hyper-rectangles, referred to as regions and denoted $\mathcal{R}_k = \prod_{\ell=1}^p [a_{k,\ell}, b_{k,\ell}], \ \forall k, 1 \le k \le K$ obtained by dyadic splits[2]. A weight $\gamma_k$ is associated to the $k$-th region $\mathcal{R}_k$ leading, for observations $\mathbf{x} \in \mathbb{R}^p$, to a predictor of the form: $f(\mathbf{x}; \Theta) = \sum_{k=1}^K \gamma_k \mathbb{1}_{\{\mathbf{x} \in \mathcal{R}_k\}}$.

As noted in previous studies [14, 20], regression trees, as they are based on constant piece-wise functions, may fail to accommodate the smoothness of the link function. To solve this problem, we introduce a prediction function that generalizes the one of standard regression trees by replacing its indicator function with a smooth function $\Psi$:

$$T_{\text{PR}}(\mathbf{x}; \Theta) = \sum_{k=1}^K \gamma_k \Psi(\mathbf{x}; \mathcal{R}_k, \boldsymbol{\sigma}). \tag{2}$$

The set of parameters $\Theta = ((\mathcal{R}_k)_{1 \le k \le K}, \boldsymbol{\gamma}, \boldsymbol{\sigma})$ corresponds to the set of regions, associated weights represented by $\boldsymbol{\gamma} \in \mathbb{R}^K$ and noise in the input variables captured in $\boldsymbol{\sigma} \in \mathbb{R}_+^p$. In the remainder, we will refer to $\boldsymbol{\sigma}$ as the *noise vector*. Note that when $\Psi(\mathbf{x}; \mathcal{R}_k, \boldsymbol{\sigma}) = \mathbb{1}_{\{\mathbf{x} \in \mathcal{R}_k\}}, \forall k, 1 \le k \le K$, one recovers standard regression trees, and so the consideration of $\boldsymbol{\sigma}$ extends the standard framework for regression trees, which does not explicitly take into account the fact that the input may be noisy.

The functions $\Psi$ we consider relate, through a *sufficiently regular* probability density function $\phi$ ($\phi$ is in $L^2$, is of class $\mathcal{C}^1$ and the support of its Fourier transform is $\mathbb{R}^p$), data points to different regions of the tree and smooth the predictions made. They are defined, for all $\mathbf{x} \in \mathcal{X}$, by:

$$\Psi(\mathbf{x}; \mathcal{R}_k, \boldsymbol{\sigma}) = \frac{1}{\prod_{j=1}^p \sigma_j} \int_{\mathcal{R}_k} \phi\left( \left( \frac{u_j - x_j}{\sigma_j} \right)_{1 \le j \le p} \right) d\mathbf{u}. \tag{3}$$

In practice, experts may have an empirical knowledge on the nature of the errors (for example when measurements are done by calibrated machines) which can help to choose $\phi$. We will make use of

different distributions on different datasets in our experiments. As one can expect, the best choice for $\phi$, which can be established via cross-validation, depends on the collection considered. Standard regression trees are obtained by considering a Gaussian distribution for $\phi$ with $\sigma_j \to 0, \forall j, 1 \leq j \leq p$. In that case, the distribution of $\mathbf{x}$ over regions is concentrated on one region.

**Parameter estimation**   Given a training set $\mathcal{D}_n = \left\{ (\mathbf{x}^{(i)}, y^{(i)})_{1 \leq i \leq n} \right\}$, with $\mathbf{x} \in \mathbb{R}^p$, $y \in \mathbb{R}$, and in accordance with the empirical risk minimization principle with a quadratic loss, the estimation procedure for probabilistic regression trees aims at finding the parameters $\Theta$ solutions of:

$$\underset{\Theta}{\operatorname{argmin}} \sum_{i=1}^{n} \left( y^{(i)} - \sum_{k=1}^{K} \gamma_k P_{ik} \right)^2, \text{ with } P_{ik} := \Psi(\mathbf{x}^{(i)}; \mathcal{R}_k, \boldsymbol{\sigma}). \tag{4}$$

The $n \times K$ matrix $\boldsymbol{P} = (P_{ik})$ thus encodes the relations between each training example $\mathbf{x}^{(i)}$ and each region $\mathcal{R}_k$. It is such that $0 \leq P_{ik} \leq 1$ and $\forall i, 1 \leq i \leq n, \sum_{k=1}^{K} P_{ik} = 1$.

The estimation of the different parameters in $\Theta$ alternates in between region and weight estimates, as in standard regression trees, till a stopping criterion is met[3]. During this process, the number of regions is increased and the matrix $\boldsymbol{P}$ and the weights $\boldsymbol{\gamma}$ are gradually updated.

*Estimating $\boldsymbol{\gamma}$* – When fixing the regions $(\mathcal{R}_k)_{1 \leq k \leq K}$ and the vector $\boldsymbol{\sigma}$, minimizing Eq. (4) with respect to $\boldsymbol{\gamma}$ leads to the least square estimator defined by $\hat{\boldsymbol{\gamma}} = \left( \boldsymbol{P}^T \boldsymbol{P} \right)^{-1} \boldsymbol{P}^T \boldsymbol{y}$ if $\mathbf{P}^T \mathbf{P}$ is not singular. We discuss in the Supplementary Material some conditions on the invertibility of $\mathbf{P}^T \mathbf{P}$.

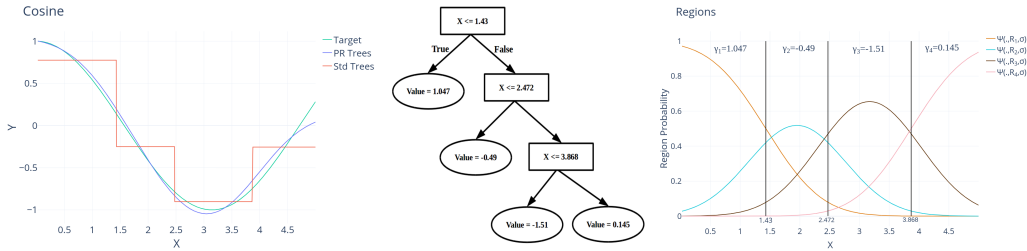

Figure 1: Left: plot of the true regression function, the probabilistic regression tree prediction and the standard regression tree prediction. Middle: PR tree learned from 100 observations where the stopping criteria consists in having at least $20\%$ of the observations in each leaf. Right: description of the probabilistic regression tree parameters: regions (corresponding to intervals as $p = 1$), $\boldsymbol{\gamma}$ in each region and the functions $\Psi$ associated to each region.

*Estimating $(\mathcal{R}_k)_{1 \leq k \leq K}$* – Let us assume that $K$ regions, referred to as *current regions*, have already been identified, meaning that the current tree has $K$ leaves. As in standard regression trees, each current region $\mathcal{R}_k$, $1 \leq k \leq K$, can be decomposed into two sub-regions with respect to a coordinate $1 \leq j \leq p$ and a splitting point $s_k^j$ that minimizes Eq. (4). Each split leads to the update of $\mathbf{P}$, that now belongs to $M_{n,K+1}(\mathbb{R})$, and $\boldsymbol{\gamma}$, that now belongs to $\mathbb{R}^{K+1}$. Substituting $\boldsymbol{\gamma}$ by its value, the best split for the current region $\mathcal{R}_k$ is given by:

$$\underset{1 \leq j \leq p, s \in \mathcal{S}_k^j}{\operatorname{argmin}} \sum_{i=1}^{n} \left( y^{(i)} - \sum_{l=1}^{K+1} \left( \left( \boldsymbol{P}^T \boldsymbol{P} \right)^{-1} \boldsymbol{P}^T \boldsymbol{y} \right)_l P_{il} \right)^2, \tag{5}$$

where $\mathcal{S}_k^j$ denotes the set of splitting points for region $\mathcal{R}_k$ and variable $j$ (more precisely, $\mathcal{S}_k^j$ is the set of middle points of the observations from $\mathcal{R}_k$ projected on the $j$th coordinate).

*Estimating $\boldsymbol{\sigma}$* – Lastly, the vector $\boldsymbol{\sigma}$ can either be based on *a priori* knowledge or be learned through a grid search on a validation set. We rely on the latter in our experiments.

**Illustration**   As an illustration, we consider a toy example based on $Y = \cos(X) + \varepsilon$, with $X \sim \mathcal{U}([0, 5])$ and $\varepsilon \sim \mathcal{N}(0, 0.05^2)$. A training set of 100 observations is generated from this model.

For the PR tree, the function $\Psi$ is set to the Gaussian distribution[4], for $1 \leq k \leq K$,

$$\Psi(\mathbf{x}; \mathcal{R}_k, \sigma) = \frac{1}{\sigma\sqrt{2\pi}} \int_{a_k}^{b_k} e^{-\frac{(u-x)^2}{2\sigma^2}} du. \tag{6}$$

where the variance is fixed to $\sigma^2 = \widehat{\mathrm{var}}((x^{(i)})_{1 \leq i \leq n})/2 = 0.74^2$. Figure 1 (left) compares the performance of standard tree and Probabilistic Regression tree to the true regression function. Unlike the standard regression tree, which is based on constant piece-wise functions, the PR tree, given in Figure 1 (middle), is able to accurately approximate the true function, even though few observations are available. Lastly, Figure 1 (right) illustrates the weights $\gamma$ as well as the distribution of $\Psi$ across the regions. As $\sigma^2$ is relatively large in this case, there is a large overlap between the functions $\Psi$ across regions, leading to a smooth prediction function.

**Consistency** We show here that the PR tree learned from a training set of size $n$, denoted $\hat{T}_{\mathrm{PR}}^{(n)}$, is consistent, that is $\lim_{n \to +\infty} \mathbb{E}[|\hat{T}_{\mathrm{PR}}^{(n)}(\mathbf{X}) - \mathbb{E}(Y|\mathbf{X})|^2] = 0$. All the proofs, which are partly based on the density results described in Schaback [24], Devore and Ron [7] and the consistency results established in Györfi et al. [13], Scornet et al. [25], are fully detailed in the Supplementary Material.

To emphasize the fact that the trees learned depend on a training set of size $n$, we denote, in this section as well as in the Supplementary Material, the $k$-th region as $\mathcal{R}_k^{(n)}$ with $k = 1, \cdots, K_n$. In the sequel, we assume that the sequence $(K_n)$ is non decreasing. Furthermore, for the sake of simplicity, we consider that the observations lie in $[0,1]^p$, the extension to any compact subspace $\mathcal{X}$ of $\mathbb{R}^p$ being direct. We also denote the Fourier transform of any function $f$ as $\mathcal{F}f$ and assume that the noise vector $\boldsymbol{\sigma}$ is fixed and known[5].

The function $\Psi$ at the basis of PR trees (Eq. (3)) relies on a probability density function $\phi$ that regulates how each data point is distributed across all regions of the tree. We assume here that $\phi$ satisfies the following conditions: the support of its Fourier transform is $\mathbb{R}^p$ ; $\exists r > 0$, $\sup_{\mathbf{v} \in \mathbb{R}^p} |\mathbf{v}|^{1+r+p/2}|\phi(\mathbf{v})| < \infty$ (which ensures that $\phi \in L^2$) ; and $\phi \in \mathcal{C}^1$.

The estimators provided by PR trees belong to the function space $\mathcal{V}_n$ defined by:

$$\mathcal{V}_n = \left\{ \sum_{k=1}^{K_n} A_k \Psi(\cdot; \mathcal{R}_k^{(n)}, \boldsymbol{\sigma}), (A_k) \in \mathbb{R}^{K_n} \right\}.$$

As usual in consistency theory [13], we first focus on truncated versions of the different functions. From the remainder of this section, let $(\beta_n)$ be a sequence of non negative numbers such that $\lim_{n \to +\infty} \beta_n = +\infty$. We define $\widetilde{T}_{\mathrm{PR}}^{(n)} = \mathcal{T}_{\beta_n} \hat{T}_{\mathrm{PR}}^{(n)}$ the truncated estimator as:

$$\mathcal{T}_{\beta_n} u = \left\{ \begin{array}{c} u \text{ if } |u| < \beta_n, \\ \mathrm{sign}(u)\beta_n \text{ if } |u| \geq \beta_n. \end{array} \right.$$

We also define $\mathcal{T}_{\beta_n}\mathcal{V}_n = \{g \in \mathcal{V}_n : \|g\|_\infty \leq \beta_n\}$, and consider the Sobolev space of functions defined, for $s \in (1,2)$, by $H^s([0,1]^p) = \{f \in L^2([0,1]^p), \exists g \in H^s(\mathbb{R}^p) \text{ s.t. } f = g|_{[0,1]^p}\}$, with $H^s(\mathbb{R}^p) = \{f \in L^2(\mathbb{R}^p), (1 + ||\cdot||_2^2)^{\frac{s}{2}}|\mathcal{F}f(\cdot)| \in L^2(\mathbb{R}^p)\}$. Note that $H^s([0,1]^p)$, that contains functions from $L^2$ for which the derivative of order $s$ is also in $L^2$, encompasses many usual functions.

As detailed in the Supplementary Material, $\mathcal{T}_{\beta_n}\mathcal{V}_n$ is dense in $H^s([0,1]^p)$:

**Proposition 1.** *Assume that for some $s \in (1,2)$, $\mathbb{E}(Y|\mathbf{X} = \cdot) \in H^s([0,1]^p)$. Let $(\lambda_n)$ be a non decreasing sequence such that for $n$ sufficiently large ,*

$$\beta_n > \frac{\lambda_n^{p/2} \|\mathbb{E}(Y|\mathbf{X} = \cdot)\|_{H^s([0,1]^p)}}{\inf_{|\boldsymbol{\omega}| \leq \lambda_n} |\mathcal{F}\phi(\boldsymbol{\omega})|}. \tag{7}$$

*Then, one has for an absolute constant C and for any $M > 0$:*

$$\inf_{g \in \mathcal{T}_{\beta_n} \mathcal{V}_n} \|\mathbb{E}(Y|\mathbf{X} = \cdot) - g\|^2_{L^2([0,1]^p)} \leq C\|\mathbb{E}(Y|\mathbf{X} = \cdot)\|^2_{H^s([0,1]^p)} \, a_n, \quad (8)$$

*with* $a_n = \left( \dfrac{1}{(1 + |\lambda_n|^2)^{s/4}} + \beta_n M^{-r} + \beta_n \left[ \max_{k=1,\dots,K_n} \text{diam}(\mathcal{R}^{(n)}_k \cap [-M, M]^p) \right] \right)^2.$

Note that in whole generality, our regions $\mathcal{R}^{(n)}_k$ may be *unbounded*, whereas we want to approximate our unknown link function $\mathbb{E}(Y|\mathbf{X} = \cdot)$ on *the compact* set where lie the observations. Proposition 1 states that, up to a controlled error $\beta_n M^{-r}$, we can replace each region by its restriction to a given compact $[-M, M]^p$.

For the estimation error, one can first bound the difference between the empirical and *real* errors for functions in $\mathcal{T}_{\beta_n} \mathcal{V}_n$ through:

**Proposition 2.** *Let* $(\mathbf{X}^{(1)}, Y^{(1)}), \cdots, (\mathbf{X}^{(n)}, Y^{(n)})$ *be i.i.d. copies of the vector* $(\mathbf{X}, Y)$. *For any* $n \in \mathbb{N}$, *any* $g \in \mathcal{T}_{\beta_n} \mathcal{V}_n$ *and for any* $\eta > 0$,

$$\mathbb{E}\left[ \sup_{g \in \mathcal{T}_{\beta_n}(\mathcal{V}_n)} \left| \frac{1}{n} \sum_{i=1}^{n} |g(\mathbf{X}^{(i)}) - \mathcal{T}_{\beta_n} Y^{(i)}|^2 - \mathbb{E}[|g(\mathbf{X}) - \mathcal{T}_{\beta_n} Y|^2] \right| \right] \leq \eta + 64\beta_n^2 \exp[-c_n],$$

$$\text{with } c_n = \frac{n}{\beta_n^4} \left\{ \frac{\eta^2}{2048} - \frac{\beta_n^4 (K_n - 1) \log(pn)}{n} - \frac{2\beta_n^4 K_n}{n} \log\left( \frac{C'\beta_n^2}{\eta} \right) \right\},$$

*and where* $C' > 0$ *is a universal constant.*

Let us now set $\beta_n = \|\mathbb{E}(Y|\mathbf{X} = \cdot)\|_\infty + \tilde{\sigma}\sqrt{2}(\log n)^2$. Consider $\lambda_n$ such that Eq (7) is satisfied. Moreover, assume that: $K_n \xrightarrow[n \to +\infty]{} +\infty$, and $K_n(\log n)^9/n \xrightarrow[n \to +\infty]{} 0$, which means that the number of regions needed to explain an infinite number of data points is also infinite, however still largely (by a factor $(\log n)^9$) dominated by the number of points. Then the above results lead to the consistency of the truncated estimator $\widetilde{T}^{(n)}_{\text{PR}}$. The extension to the consistency of the untruncated estimator $\hat{T}^{(n)}_{\text{PR}}$ follows from $\lim_{n \to +\infty} \mathbb{E}[|\hat{T}^{(n)}_{\text{PR}}(\mathbf{X}) - \tilde{T}^{(n)}_{\text{PR}}(\mathbf{X})|^2] = 0$, derived from arguments similar to the ones developed in Scornet et al. [25]. This finally leads to:

**Theorem 1.** *Set* $M > 0$. *With* $K_n$ *as above, assume that for some* $s \in (1, 2)$, $\mathbb{E}(Y|\mathbf{X} = \cdot) \in H^s([0, 1]^p)$ *and that* $\max_{k=1,\cdots,K_n} \left[ \text{diam}(\mathcal{R}^{(n)}_k \cap [-M, M]^p) \right] \xrightarrow[n \to +\infty]{} 0$. *Then*

$$\lim_{n \to +\infty} \mathbb{E}[|\hat{T}^{(n)}_{PR}(\mathbf{X}) - \mathbb{E}(Y|\mathbf{X})|^2] = 0.$$

The assumption on the diameter of the regions is reasonable for data points lying in a compact subspace: as the number of regions grows (to infinity) with the number of data points, their diameter will decrease, data points being more and more concentrated in each region. An important difference with respect to Biau et al. [1], Scornet et al. [25] is that we are not averaging over independent classifiers as regions are dependent on each other. Lastly, note that the only assumption made on the link function is that it belongs to a particular Sobolev space; no modeling assumption, as the additive assumption made in Scornet et al. [25], is required here.

## 4 Experimental Validation

We illustrate in this section the behavior of PR trees on several collections, and compare them with standard regression trees, Soft regression trees [14] and STR trees[6].

*Methods* For Soft trees, we use the implementation available in github[6] with the default parameters. For STR trees and BooST, its extension to GBT, we use the implementation available in github[7]. For standard regression trees and their ensemble extensions, we use the implementation from Scikit-Learn [29]. PR trees are built on top of this implementation, and the code for this implementation is

available in https://gitlab.com/sami.kh/pr-tree. To compute the weights $\gamma$, we rely on the Moore-Penrose pseudo-inverse. In addition, unless otherwise specified, we use the Normal distribution for $\Psi$ (Eq. 3). Lastly, for both PR and standard regression trees, the stopping criterion is the same in all experiments: all leaves should contain at least 10% of the training data. The stopping criterion for STR trees is based on the number of regions, which is chosen to be equal to the ones obtained for PR and standard trees. For Soft trees, the growth is stopped when the performance on a validation set decreases, which gives them a slight advantage in terms of performance. All the results are evaluated using the Root Mean Squared Error (RMSE).

*Complexity* As the complexity for training PR trees is $\mathcal{O}(pSK^2n)$, where $S$ is a constant upper bounding the number of considered splitting points (see the Supplementary Material), we consider a subset of splitting variables consisting of the top $V$ variables according to the splitting criterion of standard regression trees (this last step is negligible when the tree is of depth 2 or more). This finally leads to an overall complexity of $\mathcal{O}(VSK^2n)$. In the remainder, $V$ is set to 3. Note that the results obtained with this and higher values are close to the ones obtained when considering all variables (see the Supplementary Material).

*Data sets* We make use here of 13 data sets of various size, namely (ordered by increasing sample size) Riboflavin (RI), Ozone (OZ), Diabetes (DI), Abalone (AB), Boston (BO), Bike-Day (BD), E2006, Skill (SK), Ailerons (AL), Bike-Hour (BH), Super Conductor (SC), Facebook Comments (FC) and Video Transcoding (VT), all commonly used in regression tasks. In the experiments reported here, we use the original data sets, without any modification, to illustrate the fact that one can gain by treating *real* data sets as noisy. Full details on these datasets (sample size and number of variables, location) are given in the Supplementary Material.

*Experiments* We use stratified 10-fold cross-validation to evaluate the performance of each method. Each fold is divided into 80% for train and 20% for test, except for Soft trees and PR trees and their gradient boosted extension (see below) for which each fold is divided into 65% for train, 15% for validation and 20% for test. For Soft trees, the validation set is used for the stopping criterion. For PR trees and their gradient boosted extension, it is used to estimate the noise vector $\boldsymbol{\sigma}$ through a grid search taking values, for each variable $j$, $1 \leq j \leq p$, in the interval $[0, 2\hat{\sigma}_j]$ with a step of $\hat{\sigma}_j/4$, where $\hat{\sigma}_j$ denotes the empirical standard deviation of variable $j$. All experiments were conducted on a 256 GB RAM server with 32 CPUs at 2.60GHz. Lastly, a two-sided t-test at the level 0.05 is used to assess whether differences are significant or not.

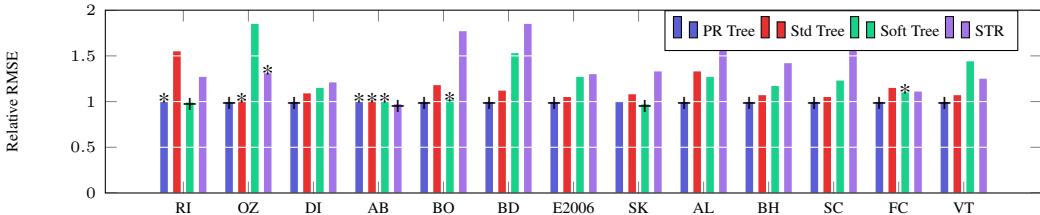

Figure 2: Normalized results obtained with 10 stratified cross-validation on PR, standard, Soft and STR trees. '+' means the best result, and '*' corresponds to results that are not significantly worst than the best one according to a two-sided t-test at 5%.

**Results for tree-based methods** Figure 2 displays the results obtained with the different regression trees (full performances stand in Supplementary Material). As one can note, PR trees significantly outperform standard, Soft and STR trees, on almost all data sets. Standard regression trees never significantly outperform PR trees whereas Soft trees are only significantly better than PR trees on one collection. In contrast, PR trees are significantly better than Soft trees on eight collections and significantly better than standard and STR trees on eleven collections. When the number of examples is sufficiently large, *e.g.*, on datasets containing more than 5,000 examples (as AL, BH, SC, FC, VT), the difference between PR and Soft and STR trees is more important. This indicates that the models learned by Soft and STR trees, which are more complex as they rely on more parameters, have a tendency to overfit in this case. This indicates that the models learned by Soft and STR trees, which are more complex as they rely on more parameters, have a tendency to overfit in this case. Lastly, when the number of observations is small compared to the number of features, as on RI and E2006

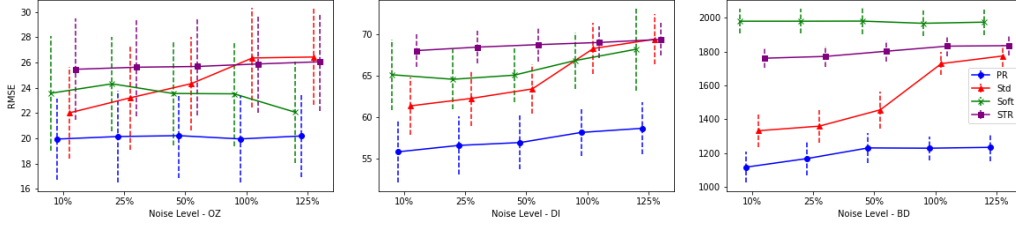

Figure 3: RMSE for the different trees across 5 noise levels on OZ (left), DI (middle) and BD (right).

which resp. contain 71 and 3308 examples for 4088 and 9000 features, PR trees overall outperform the other methods. On RI, with very few examples, PR trees, despite the fact that the validation to select the noise vector is done on very few examples, and Soft trees yield the best results. On E2006, PR trees significantly outperform the other methods, in line with the previous remark.

**Interpretability, adaptability and robustness**    As mentioned before, PR trees can be easily interpreted. In addition, they are adaptable in the sense that different distribution functions can be used to smooth the prediction, and they are relatively robust to noise. We illustrate these points below and refer the reader to the Supplementary Material for more details.

*Interpretability* To illustrate how observations are linked to the different regions, one can see in the figure on the right, for OZ, DI and BD, the boxplots for all observations of the probability to belong to the three most probable regions ($K_1^*$ denotes the most probable region for any observation, $K_2^*$ the second most probable region and $K_3^*$ the third most probable region). For the three data sets and for most observations, roughly 75% of their distribution is concentrated on these three regions, which are by far the most important ones and can be used to provide a first explanation for the values predicted.

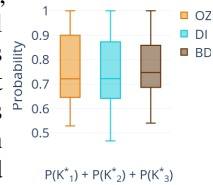

*Adpatability to the underlying probability distribution* Another advantage of PR trees is that different distribution functions can be used to smooth the prediction (function $\phi$ in Eq. 3). The choice of $\phi$ can be made according to some *a priori* knowledge on the nature of the errors or by testing different distributions and selecting the best one on a validation set. We provide in the Supplementary Material the results of an experiment conducted on seven datasets with six different distributions (two variants of the Gamma distribution, the Laplace distribution, the Lognormal distribution, the Normal distribution and two variants of the Student distribution). As one can expect, the choice of the best distribution depends on the collection, the Student distribution with 3 degrees of freedom being, for example, particularly adapted to the AL, BO and DI collections. This said, the Normal distribution behaves well on all collections, hence its choice in our experiments.

*Robustness* Because they allow data points to be probabilistically assigned to different regions, PR trees are more robust to noise than their standard counterpart. To illustrate this point, we added, for each variable $j$, $1 \leq j \leq p$, some random Gaussian noise $\mathcal{N}(0, \sigma_{\tau,j}^2)$ with $\sigma_{\tau,j} = (\tau \hat{\sigma}_j)_{1 \leq j \leq p}$, $\tau \in \{0.1, 0.25, 0.5, 1, 1.25\}$, corresponding to 5 noise levels on the empirical standard deviation $\hat{\sigma}_j$. We then evaluated how the RMSE evolves according to the noise level on three collections, OZ, DI and BD. The results are summarized in Figure 3. As one can note, PR trees are more robust to noise than standard trees, and to a lesser extent than Soft trees. STR trees are relatively stable as well (as Soft trees on two datasets) but with worse performance, limiting their interest.

**Ensemble extensions**    The extension of PR trees to Random Forests, denoted by PR-RF is straightforward. Note that the noise vector is fixed to the one obtained for a single tree as each tree aims at predicting the output variable $Y$. On all the collections described above, we ran both PR and standard Random Forests with 100 trees. The default parameters (consisting of all variables with a sampling strategy consisting of bootstrap with replacement) are used in both cases. The results obtained (presented in the Supplementary Material) show that PR-RF significantly outperform their standard counterpart on seven collections (out of 13), are on a par on five collections and are significantly worse on only one collection. Random Forests aim at reducing the variance (at the expense of a small increase in the bias), whereas our adaptation to smoothness aims at reducing the bias. Combining both,

as in PR-RF, reduces both bias and variance and leads to a method which significantly outperforms RF.

For using PR trees in Gradient Boosted Trees, an extension we refer to as PR-GBT, assuming that $(T-1)$ trees have been built so far, we simply replace $y^{(i)}$ in Eq. 4 by its residual $(y^{(i)} - \sum_{t=1}^{T-1} \sum_{k=1}^{K_t} \gamma_k P_{ik})$. One can directly use the procedure given in Section 3 to estimate the new tree in the ensemble. However, this time, the noise vector may change from one tree to another as a different output is predicted. We thus tune the noise vector on each tree independently, using the same grid search as the one detailed in the previous section. Figure 4 compares PR-GBT with standard Gradient Boosted Trees, denoted by GBT, and BoosT, the extension of STR trees to Gradient Boosted Trees. As one can note, PR-GBT is the best performing method overall, even though all methods are on a par on five collections and BoosT obtains results close to PR-GBT (the latter being significantly better than the former on 4 collections, and significantly worse on 3 collections).

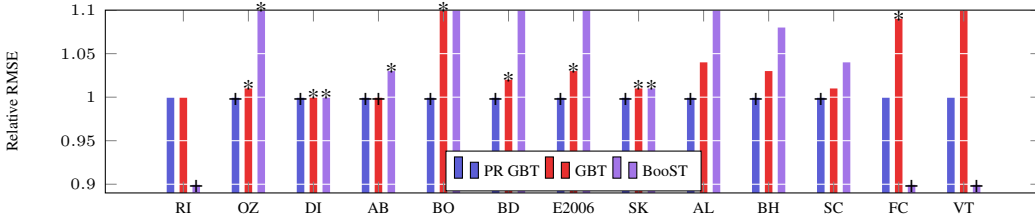

Figure 4: Normalized results obtained with 10 stratified cross-validation on PR GBT, standard GBT and BoosT. '+' means the best result, and '*' corresponds to results are not significantly worst than the best one, according to a t-test at $5\%$. Values above 1.1 and below 0.9 have been truncated.

## 5   Conclusion

We have studied here approximations, based on regression trees, of smooth link functions. Standard regression trees, while being a basic building block for two of the most popular and efficient ensemble methods, Random Forests and Gradient Boosted Trees, are based on constant piece-wise functions and may fail to accommodate the smoothness of the link function. To solve this problem, we have introduced functions that relate, through sufficiently regular probability density functions, data points to different regions of the tree and smooth the predictions made. We have then shown that the probabilistic regression trees thus obtained are consistent, *i.e.*, $\lim_{n\to+\infty} \mathbb{E}[\hat{T}_s^{(n)}(\mathbf{X}) - \mathbb{E}(Y|\mathbf{X})]^2 = 0$, with $\hat{T}_s^{(n)}$ the probabilistic regression tree learned from a training set of size $n$, and outperform, on a variety of collections, previously proposed trees (as standard, Soft and Smooth Transition Regression trees) in terms of performance and robustness to noise. Lastly, we have proposed versions of Random Forests and Gradient Boosted Trees based on Probabilistic Regression trees and shown that these versions outperform the state-of-the-art.

In the future, we want to extend the consistency results of probabilistic regression trees to their ensemble extensions, and derive Bayesian versions of these extensions, following the work conducted in Chipman et al. [5] and Linero and Yang [20]. We also plan to investigate knowledge distillation, as introduced in Frosst and Hinton [12].

## Broader Impact

This work is mainly theoretical, with standard regression applications.

## Acknowledgments and Disclosure of Funding

This work has been partially supported by MIAI@Grenoble Alpes (ANR-19-P3IA-0003) and by the french PIA project *Lorraine Université d'Excellence* (ANR-15-IDEX-04-LUE).

## Footnotes

[1]*The State of Data Science & Machine Learning*; https://www.kaggle.com/surveys/2017

[2]For $1 \le k \le K$, $(a_{k,j}, b_{k,j}) \in (-\infty, +\infty)^2$ and the segments are extended in the obvious way when $a_{k,j} = \pm\infty$ or $b_{k,j} = \pm\infty$, $1 \le j \le p$.

[3]Any standard stopping criterion, as tree depth or number of examples in a leaf, can be used here.

[4]Note that in this case, defining the distance between an observation $\mathbf{x}$ and a region $\mathcal{R}_k$ by $d(\mathbf{x}, \mathcal{R}_k) = \inf_{\mathbf{z} \in \mathcal{R}_k} ||\mathbf{x} - \mathbf{z}||_2^2$, the closer $\mathbf{x}$ is to region $\mathcal{R}_k$, the most important the contribution of $\mathcal{R}_k$ is in the prediction of $Y$ given $\mathbf{x}$.

[5]Note that this assumption is in line with the procedure we rely on to estimate $\boldsymbol{\sigma}$.

[6]https://github.com/oir/soft-tree

[7]https://github.com/gabrielrvsc/BooST

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
