[Supplementary Material 1]

# Supplementary Material for
## *Smooth And Consistent Probabilistic Regression Trees*

We provide in this supplementary file detailed results for the experiments (Section A), a sufficient condition for the invertibility of $\mathbf{P}^T\mathbf{P}$ (Section B), and finally the proof for the consistency of probabilistic regression trees (Section C).

## Contents

# A   Experiments

## A.1   Datasets

We present in this section the datasets used in Section 4 of the main paper. We perform our experiments on several classical dasets of the litterature, namely Abalone[1][2] (AB) , which was used in [Meinshausen, 2006], Riboflavin [1](RI) which was used in [Bühlmann et al., 2014] as well as Ailerons[2] (AL), Bike-Day[1] (BD),

Bike-Hour[1] (BH), Boston[3] (BO), Diabetes[14] (DI), Facebook Comments[1] (FC), Forest Fires[1] (FF), Ozone[5] (OZ), E2006[6], Skill (SK), Super Conductor[1] (SC) and Video Transcoding[7] (VT). Table 6 below provides the main characteristics of these datasets.

Table 1: Benchmark data sets with their characteristics, sorted by the sample size.

| Data set | RI | OZ | DI | AB | BO | BD | E2006 | SK | AL | BH | SC | FC | VT |
|---|---|---|---|---|---|---|---|---|---|---|---|---|---|
| Features | 4088 | 10 | 10 | 7 | 11 | 16 | 9000 | 18 | 40 | 16 | 81 | 40 | 20 |
| Samples | 71 | 112 | 441 | 500 | 505 | 730 | 3308 | 3337 | 7154 | 17389 | 21263 | 40949 | 68785 |

## A.2   Computational considerations

For the construction of the probabilistic regression tree, the most time consuming operation is the one that determines the best split to be taken at each current leaf. The computation of $\mathbf{P}$ can be done incrementally, with a complexity of $\mathcal{O}(n)$ each time a node is subdivided into two nodes. As the number of leaves is usually small compared to the number of training examples, the computation of $(\mathbf{P}^T\mathbf{P})^{-1}\mathbf{P}^T$ requires $\mathcal{O}(K^2 n)$ operations, if $K$ regions have already been constructed. Let $\mathcal{S}_k = \cup_{j,\, 1 \le j \le p} \mathcal{S}_k^j$ (where $\mathcal{S}_k^j$ denotes as before the set of splitting points for region $\mathcal{R}_k$ and for variable $j$) and let us assume that, $\forall k, 1 \le k \le K, |\mathcal{S}_k| < S$ where $S$ is a constant. Note that one can always define $S$ by taking the maximum number of potential split points for all variables. Then, the overall complexity for computing the best split is $\mathcal{O}(pSK^2 n)$. This contrasts with the complexity for finding the best split in a given region in standard regression trees that only amounts to $\mathcal{O}(pS\mathcal{D}_{n,k})$ where $\mathcal{D}_{n,k}$ denotes the number of training examples in the region.

Note that the computational complexity of probabilistic regression trees is linear in $p$, the number of features; they can thus be used on datasets for which $p$ is large. This said, in practice, one can be faster by considering a smaller number, $V$, of potential variables to split. We consider the top $V$ variables according to the splitting criterion of standard regression trees (this last step is negligible when the tree is of depth 2 or more). This leads to an overall complexity of $\mathcal{O}(VSK^2 n)$, where typical values of $V$ are 1, 3 and 5. In the experiments, we have seen no significant difference between considering all variables (V=p) or only three (V=3). Table 2 summarizes the results.

Table 2: Difference in prediction (RMSE, mean and sd in parenthesis) between $V = p$ and $V = 3$.

| Dataset | AB | BD | BO | DI | OZ |
|---|---|---|---|---|---|
| Top V=3 | 3.11(0.27) | 374.2(43.2) | 4.47(1.04) | 55.92(3.97) | 18.66(3.65) |
| Top V=p | 3.1(0.28) | 373.6(43.86) | 4.15(0.57) | 55.73(4.17) | 18.69(3.75) |

For prediction, one needs to compute $\Psi(\mathbf{x}; \mathcal{R}_k, \boldsymbol{\sigma})$, which can be done, for each region, in $\mathcal{O}(p)$ operations using a standard normal table. The overall time complexity for prediction is thus $\mathcal{O}(Kp)$.

Table 3: Time computation in seconds.

| Dataset | PR V=3 | Soft Tree | BooST | MARS | Std Tree |
|---|---|---|---|---|---|
| BD | 5,68 | 5,75 | 0,24 | 0,17 | 0,00065 |
| BO | 3,19 | 14,41 | 0,20 | 0,27 | 0,0007507901 |
| DI | 2,75 | 0,73 | 0,19 | 0,17 | 0,0005473019 |
| SC | 411,43 | 32087 | 33,91 | 212,93 | 0,2108744748 |

## A.3 Results for tree-based methods

We add details about the results given in Figure 2 of the experimental results of Section 4. In particular, we give in Table 6 the full results related to the performance of the different regression tree state-of-art methods, that we compare to PR Tree. We added a '*' subscript, to indicate when a method significantly outperforms the others. In this table, we keep 3 values for $V$ the number of top variables to select in the PR tree. It highlights that even if there is no general rule, $V = 3$ has the best results for a reasonable computation time. We will keep $V = 3$ in the other experiments.

Table 4: Results for one tree obtained with 10 stratified cross-validation on PR, standard, Soft and STR trees. '*' corresponds to the best result, and bold represents results that are not significantly different than the best one according to a two-sided t-test at 5%..

| Dataset | PR Tree V=1 | PR Tree V=3 | PR Tree V=5 | Standard Tree | Soft Tree | STR tree M=1 |
|---|---|---|---|---|---|---|
| RI | 0.86(0.12) | **0.67(0.13)** | 0.69(0.15) | 1.04(0.15) | **0.66(0.17)*** | 0.85(0.21) |
| OZ | **17.82(2.4)*** | **18.66(3.65)** | **18.88(3.23)** | **18.9(3.39)** | **34.44(43.27)** | 24.5(3.58) |
| DI | **57.05(3.82)** | **55.92(3.97)** | **55.76(3.96)*** | 60.95(3.92) | 64.18(4.15) | 67.89(1.99) |
| AB | **3.08(0.29)** | **3.11(0.27)** | **3.08(0.28)** | **3.15(0.29)** | **3.11(0.23)** | **3.03(0.24)*** |
| BO | 4.7(0.88) | **4.47(1.04)** | **4.21(0.88)*** | 5.27(0.61) | 4.54(0.97) | 7.91(0.42) |
| BD | **925.18(70.79)** | **898.11(55.43)*** | **918.46(47.95)** | 1006.73(52.07) | 1376.74(165.24) | 1661.4(54.23) |
| E2006 | **0.37(0.02)** | **0.37(0.02)*** | **0.37(0.02)** | 0.39(0.02) | 0.47(0.02) | 0.48(0.02) |
| SK | 1(0.04) | 0.98(0.05) | 0.98(0.03) | 1.06(0.03) | **0.95(0.02)*** | 1.3(0.03) |
| AL | **1.23(0.03)*** | **1.24(0.02)** | **1.24(0.02)** | 1.65(0.03) | 1.57(0.7) | 2.28(0.03) |
| BH | 128.12(1.29) | 128.52(1.84) | **126.69(1.1)*** | 128.12(1.29) | 148.63(33.42) | 169.67(1.54) |
| SC | 18.79(0.21) | **17.89(0.28)*** | 18.1(0.37) | 18.79(0.21) | 21.96(1.27) | 29.58(0.16) |
| FC | **28.77(3.32)*** | **28.87(3.32)** | **28.9(3.35)** | 33.12(3.82) | **31.68(3.47)** | **31.94(3.5)** |
| VT | 11.44(0.16) | **11.08(0.18)*** | 11.16(0.19) | 11.87(0.17) | 15.98(0.24) | 13.82(0.23) |

## A.4 Results for ensemble methods

We now give some details about our two ensemble extensions of PR-Tree, namely PR-RF and PR-GBT extending respectively PR Tree to Random Forests and Gradient Boosting Trees. In all the experiments, we fix to $t = 100$ the number of trees considered in each ensemble method. As mentioned before, we fix the number of top variables selected to $V = 3$. As in Section A.4, we added a '*' subscript, to indicate when a method significantly outperform the other ones. Note that on one fold, BooST get very bad results for Bike, which leads to an high mean with no significance difference with the other methods.

Table 5: Results for ensemble methods with t=100.
V=3 were used for PR-RF and PR-GBT.

| Data set | PR-RF | RF | PR-GBT | GBT | BooST |
|----------|-------|-----|--------|-----|-------|
| RI | **0.64(0.16)\*** | **0.71(0.19)** | 0.68(0.13) | 0.68(0.12) | **0.52(0.09)\*** |
| OZ | **16.08(2.37)\*** | **16.28(2.59)** | **15.96(3.06)\*** | **16.11(2.65)** | **17.59(4.1)** |
| DI | **54.32(3.29)\*** | **56.53(2.61)** | **57.14(3.65)\*** | **57.33(3.72)** | **57.28(3.97)** |
| AB | **3.07(0.26)\*** | **3.09(0.27)** | **3.15(0.29)** | **3.15(0.29)** | **3.26(0.37)** |
| BO | **4.01(0.73)\*** | 4.7(0.64) | **3.4(0.6)\*** | 3.77(0.65) | **4.1(1.94)** |
| BD | **838.23(51.07)** | 907.54(42.44) | **682.95(36.58)\*** | **695.83(40.86)** | **1232.09(1587.92)** |
| E2006 | **0.36(0.02)\*** | **0.38(0.03)** | **0.36(0.02)\*** | **0.37(0.02)** | 0.635(0.395) |
| SK | **0.95(0.03)\*** | 1(0.03) | **0.9(0.03)** | **0.91(0.03)** | **0.91(0.04)** |
| AL | **1.19(0.02)\*** | 1.57(0.04) | **1.11(0.02)\*** | 1.15(0.03) | 2.28(0.03) |
| BH | 128.47(1.28) | **127.15(1.34)\*** | **82.21(0.98)\*** | 84.97(1.16) | 89.04(1.64) |
| SC | **17.72(0.3)\*** | 18.41(0.2) | **13.96(0.2)\*** | 14.07(0.17) | 14.52(0.26) |
| FC | **28.76(3.51)\*** | 32.03(3.43) | 26.61(3.47) | 29.12(3.5) | **20.85(3.31)\*** |
| VT | **11.06(0.19)\*** | 11.79(0.19) | 6.55(0.15) | 8.26(0.2) | **4.59(0.12)\*** |

## A.5 Interpretability

To illustrate how observations are linked to the different regions, we display in Fig. 1, for three data sets (Boston (BO), Ailerons (AI) and Video Transcoding (VT)) the boxplots, with respect to the observations, of $\Psi(.; \mathcal{R}_{K_1^*}, \boldsymbol{\sigma})$ (left), of $\Psi(.; \mathcal{R}_{K_1^*}, \boldsymbol{\sigma}) + \Psi(.; \mathcal{R}_{K_2^*}, \boldsymbol{\sigma})$ (middle), and $\Psi(.; \mathcal{R}_{K_1^*}, \boldsymbol{\sigma}) + \Psi(.; \mathcal{R}_{K_2^*}, \boldsymbol{\sigma}) + \Psi(.; \mathcal{R}_{K_3^*}, \boldsymbol{\sigma})$ (right), where $K_1^*$ denotes the most probable region for any observation, $K_2^*$ the second most probable region and $K_3^*$ the third most probable region. For the three data sets and for most observations, roughly 75% of their distribution is concentrated on these three regions, the most probable one concentrating more than 40% of the distribution. As the number of regions is, in all cases, equal to 8, the three most probable regions are by far the most important ones and can be used to provide a first explanation for the values predicted.

Figure 1: Boxplots for the values of $\Psi(.; \mathcal{R}_k^{(n)}, \boldsymbol{\sigma})$ on the three main regions for three datasets.

## A.6 Choice of $\phi$

A key feature of our probabilistic version of Regression Trees, Random Forest and Gradient Boosting Trees is the introduction of the distribution $\phi$ involved in our smooth prediction. In Table 6, we provide an extensive

study of the impact of the choice of $\phi$ on the performance of the method. Note that interestingly, significant differences are observed depending on the dataset. Notably, the choice of a Gaussian distribution $\phi$ is in some cases significantly outperformed by other ones.

Table 6: Several distributions

| Dataset | Gamma df=3 | Gamma df=5 | Laplace | Lognorm | Normal | Student df=3 | Student df=5 |
|---------|-----------|-----------|---------|---------|--------|--------------|--------------|
| OZ | **18.38(3.5)** | **19.17(3.9)** | **18.16(3.32)** | **17.87(2.59)** | **18.66(3.65)** | **18.53(3.43)** | **19.02(3.53)** |
| DI | **58.98(2.02)** | 61.56(3.75) | **56.01(3.73)** | 60.83(3.2) | **55.92(3.97)** | **55.88(3.97)\*** | **55.95(3.69)** |
| AB | **3.07(0.24)** | **3.04(.24)** | **3.11(0.26)** | 3.05(0.25) | **3.11(0.27)** | **3.09(0.26)** | **3.1(0.26)** |
| BO | **5.05(0.69)** | **5.24(0.56)** | **4.51(0.87)** | 5.14(0.85) | **4.47(1.04)** | **4.43(1.02)\*** | **4.43(1.04)\*** |
| BD | **400.17(31.69)** | **397.04(41.09)** | **371.16(41.29)\*** | 415.43(47.28) | **374.2(43.23)** | **372.46(42.73)** | **373.21(42.94)** |
| SK | 1.04(0.03) | 1.06(0.03) | **0.97(0.04)\*** | 1.07(0.03) | **0.98(0.05)** | **0.98(0.05)** | **0.97(0.04)\*** |
| AL | 1.63(0.04) | 1.65(0.03) | **1.25(0.02)** | 1.65(0.03) | **1.24(0.02)** | **1.23(0.02)\*** | **1.24(0.02)** |

# B  Invertibility of $\mathbf{P}^T\mathbf{P}$

We introduce here a sufficient condition for the matrix $\mathbf{P}^T\mathbf{P} \in M_{K,K}(\mathbb{R})$ to be invertible. Using the same notations as the ones in the main paper and denoting by $q_\alpha$ the $\alpha$ quantile, we first note that, provided that the uncertainties are sufficiently small, the association $\Psi(\mathbf{x}^{(i)}; \mathcal{R}_k, \boldsymbol{\sigma})$ between a training example $\mathbf{x}^{(i)}$ and the region $\mathcal{R}_k$ to which it belongs is above 0.5:

**Lemma B.1.** *Using the same notations as before, let $\mathbf{x}^{(i)} \in \mathcal{D}_n$ and $\mathcal{R}_k$ be the region to which it belongs so that $\forall j$, $1 \le j \le p$, $a_{k,j} < x_j^{(i)} < b_{k,j}$.*

$$\text{If } \forall j, \ 1 \le j \le p, \ \sigma_j < \frac{b_{k,j} - a_{k,j}}{2 q_{\frac{1+0.5^{1/p}}{2}}}, \ \text{then } P_{ik} > 0.5.$$

*Proof.* Denoting $\mathbb{P}(U_j \in [a_k^j, b_k^j]|x_j^{(i)}, \sigma_j)$ the quantity $\frac{1}{\sigma_j \sqrt{2\pi}} \int_{a_{k,j}}^{b_{k,j}} e^{-\frac{(u-x_j)^2}{2\sigma_j^2}} du$ where $1 \le j \le p$, a sufficient condition for $P_{ik} > 0.5$ is:

$$\forall j, \ \mathbb{P}(U_j \in [a_k^j, b_k^j]|x_j^{(i)}, \sigma_j) > 0.5^{1/p}.$$

The above inequality is true if:

$$\forall j, \ 1 \le j \le p, \ \mathbb{P}\left(\frac{U_j - x_j^{(i)}}{\sigma_j} < \frac{b_k^j - x_j^{(i)}}{\sigma_j} \Big| x_j^{(i)}, \sigma_j\right) - \mathbb{P}\left(\frac{U_j - x_j^{(i)}}{\sigma_j} < \frac{a_k^j - x_j^{(i)}}{\sigma_j} \Big| x_j^{(i)}, \sigma_j\right) > 0.5^{1/p}.$$

This last condition is satisfied if one has, for all $1 \le j \le p$:

$$\begin{cases} \mathbb{P}\left(\frac{U_j - x_j^{(i)}}{\sigma_j} < \frac{b_k^j - x_j^{(i)}}{\sigma_j} \Big| x_j^{(i)}, \sigma_j\right) > \frac{1 + 0.5^{1/p}}{2}, \\ \mathbb{P}\left(\frac{U_j - x_j^{(i)}}{\sigma_j} < \frac{a_k^j - x_j^{(i)}}{\sigma_j} \Big| x_j^{(i)}, \sigma_j\right) < \frac{1 - 0.5^{1/p}}{2}, \end{cases}$$

which is equivalent to:

$$\frac{a_k^j - x_j^{(i)}}{\sigma_j} < q_{\frac{1-0.5^{1/p}}{2}} \ \text{and} \ \frac{b_k^j - x_j^{(i)}}{\sigma_j} > q_{\frac{1+0.5^{1/p}}{2}}.$$

Note that $q_{\frac{1-0.5^{1/p}}{2}} = -q_{\frac{1+0.5^{1/p}}{2}} < 0$. A sufficient condition is then: for all $1 \le j \le p$,

$$\forall j, \ 1 \le j \le p, \ \sigma_j < \min_{1 \le k \le K} \left\{ \frac{\min\left(x_j^{(i)} - a_k^j, b_k^j - x_j^{(i)}\right)}{q_{\frac{1+0.5^{1/p}}{2}}} \right\}.$$

Since $a_k^j < x_j^{(i)} < b_k^j$, a more conservative condition is:

$$\forall j,\, 1 \le j \le p,\, \sigma_j < \frac{b_k^j - a_k^j}{2q_{\frac{1+0.5^{1/p}}{2}}},$$

which concludes the proof. □

We can now state the main condition for invertibility.

**Theorem B.1.** *If the following condition is satisfied:*

$$\forall j,\, 1 \le j \le p,\, \sigma_j < \frac{\min\limits_{1 \le k \le K}(b_k^j - a_k^j)}{2q_{\frac{1+0.5^{\frac{1}{p}}}{2}}},$$

*then the matrix* $\mathbf{P}^T\mathbf{P}$ *is invertible.*

To prove Theorem B.1, we prove that $P$ is of full rank. To do so, we first prove the following result:

**Proposition B.1.** *Let us assume that the condition in Lemma B.1 holds. Then, the set $I_k = \{1 \le i \le n, P_{ik} > 0.5\}$ is non-empty. Let us consider, for all $1 \le k \le K$, $i_k$ a representative of this set and let us introduce the matrix $\mathbf{Q} \in M_{K,K}(\mathbb{R})$ defined, for all $1 \le l, k \le K$, by:*

$$Q_{lk} = P_{i_l k}.$$

*The $K \times K$ matrix $\mathbf{Q}$ is invertible.*

*Proof.* We first show that $\mathbf{Q}$ is a strictly dominant diagonal matrix, *i.e.*:

$$\forall k,\, Q_{kk} > \sum_{k' \ne k} Q_{kk'}.$$

Indeed, by Lemma B.1, we know that $P_{i_k k} > 0.5$ and $k$ is the only region where it is true:

$$\sum_{l \ne k} Q_{lk} \le \sum_{l \ne k} P_{i_l k} = 1 - P_{i_k k} < 0.5 < P_{i_k k} = Q_{kk}.$$

According to Hadamard's Lemma, we know that a strictly dominant diagonal matrix is invertible, which concludes the proof. □

As $\mathbf{Q}$ is invertible, $\mathbf{P}$ is of full rank, leading to the fact that $\mathbf{P}^T\mathbf{P}$ is invertible, which proves Theorem B.1.

Note that the proofs of Lemma (B.1) and Theorem B.1 do not rely on the Gaussian assumption. The above development can thus be extended to any function $\Psi$ that can be written as the product of $p$ independent distributions. Lastly, one can interpret the above results as stating that if regions are not too small with respect to $\boldsymbol{\sigma}$, then the estimator of $\boldsymbol{\gamma}$ is well defined.

# C   Consistency of probabilistic regression trees

## C.1   Notation and assumptions

We recall that, for $s > 0$, the spaces $H^s(\mathbb{R}^p)$ are defined as

$$H^s(\mathbb{R}^p) = \{g \in L^2(\mathbb{R}^p), \boldsymbol{\omega} \mapsto (1 + |\boldsymbol{\omega}|^2)^{s/2}|\mathcal{F}(g)(\boldsymbol{\omega})| \in L^2(\mathbb{R}^p)\}$$

where we consider the Fourier transform and its inverse defined by, for any $g \in L^2(\mathbb{R}^p)$,

$$\forall \boldsymbol{\omega} \in \mathbb{R}^p, \mathcal{F}(g)(\boldsymbol{\omega}) = \int_{\mathbb{R}^p} g(\mathbf{t})e^{-i\boldsymbol{\omega}.\mathbf{t}}d\mathbf{t},$$

$$\forall \mathbf{t} \in \mathbb{R}^p, \mathcal{F}^{-1}(g)(\mathbf{t}) = \frac{1}{(2\pi)^p}\int_{\mathbb{R}^p} g(\boldsymbol{\omega})e^{+i\mathbf{t}.\boldsymbol{\omega}}d\boldsymbol{\omega}.$$

For the sake of simplicity, we consider that the observations lie in $[0,1]^p$, the extension to any compact subspace $\mathcal{X}$ of $\mathbb{R}^p$ being direct. We then assume that $\mathbf{X} \in [0,1]^p$ and that the link function is also defined on $[0,1]^p$.

A function $f$ defined on $[0,1]^p$ belongs to $H^s([0,1]^p)$ if there exists $g \in H^s(\mathbb{R}^p)$ s.t. $g|_{[0,1]^p} = f$. We define the norm on $H^s([0,1]^p)$ by

$$\|f\|_{H^s([0,1]^p)} = \inf\{\|g\|_{H^s(\mathbb{R}^p)}, g \in H^s(\mathbb{R}^p), \text{ s.t. } f = g|_{[0,1]^p}\}.$$

Since the space $H^s([0,1]^p)$ is defined as the restriction of functions of $H^s(\mathbb{R}^p)$, the proof first provides some results for functions defined on $\mathbb{R}^p$, and thereafter extend these results to functions defined on $[0,1]^p$ which leads to the consistency.

In the sequel we are given $\phi$, a probability distribution function, weighting as a probability measure the belonging of each observation to each region. To simplify notations, in this supplementary material, for $\boldsymbol{\sigma}$ fixed, we define $\phi_{\boldsymbol{\sigma}}$ as

$$\forall \mathbf{u} \in \mathbb{R}^p, \phi_{\boldsymbol{\sigma}}(\mathbf{u}) := \frac{1}{\prod_j \sigma_j}\phi\left(\frac{u_1}{\sigma_1}, \cdots, \frac{u_p}{\sigma_p}\right),$$

and set $\phi_{\boldsymbol{\sigma}}^{[2]} := \phi_{\boldsymbol{\sigma}} \star \phi_{\boldsymbol{\sigma}}$, where $\star$ denotes the convolution product.

We then assume that for any $\boldsymbol{\sigma}$, the support of its Fourier transform is $\mathbb{R}^p$: $\text{supp}(\mathcal{F}\phi_{\boldsymbol{\sigma}}) = \mathbb{R}^p$. Remark that $\phi_{\boldsymbol{\sigma}} \in L^1$, as $\phi$ is a pdf. We also assume that there exists an $r > 0$ which guides its decreasing: $\sup_{\mathbf{v} \in \mathbb{R}^p} |\mathbf{v}|^{1+r+p/2}|\phi_{\boldsymbol{\sigma}}(\mathbf{v})| < \infty$, which ensures that $\phi_{\boldsymbol{\sigma}} \in L^2$. Finally, we assume that $\phi_{\boldsymbol{\sigma}} \in \mathcal{C}^1$.

Thereafter, we introduce for all $\mathbf{x} \in \mathbb{R}^p$

$$\widetilde{\Psi}(\mathbf{x}; \mathcal{R}_k^{(n)}, \boldsymbol{\sigma}) = \int_{\mathcal{R}_k^{(n)}} \phi_{\boldsymbol{\sigma}}(\mathbf{u} - \mathbf{x})d\mathbf{u}.$$

Its restriction on $[0,1]^p$ will be denoted $\Psi(\cdot; \mathcal{R}_k^{(n)}, \boldsymbol{\sigma})$. We also define $\widetilde{\mathcal{V}}_n$ as

$$\widetilde{\mathcal{V}}_n = \left\{\sum_{k=1}^{K_n} A_k\widetilde{\Psi}(\cdot; \mathcal{R}_k^{(n)}, \boldsymbol{\sigma}), (A_k) \in \mathbb{R}^{K_n}\right\},$$

and its restriction to $[0,1]^p$ as

$$\mathcal{V}_n = \left\{\sum_{k=1}^{K_n} A_k\Psi(\cdot; \mathcal{R}_k^{(n)}, \boldsymbol{\sigma}), (A_k) \in \mathbb{R}^{K_n}\right\}.$$

To control the estimation error (Section C.3), we first focus on the truncated estimator: for $(\beta_n)$ a sequence of non negative numbers such that $\lim_{n\to+\infty}\beta_n = +\infty$, let $\widetilde{T}_s^{(n)} = \mathcal{T}_{\beta_n}\hat{T}_s^{(n)}$ be the truncated estimator defined by:

$$\mathcal{T}_{\beta_n}u = \begin{cases} u \text{ if } |u| < \beta_n, \\ \text{sign}(u)\beta_n \text{ if } |u| \geq \beta_n. \end{cases}$$

We also define $\mathcal{T}_{\beta_n}\mathcal{V}_n = \{g \in \mathcal{V}_n : \|g\|_\infty \leq \beta_n\}$.

## C.2 Proof of Proposition 1: density of $\mathcal{V}_n$ in $H^s([0,1]^p)$ and approximation error

We first provide a proposition which describes the density of $\mathcal{V}_n$ in $H^s([0,1]^p)$ with an explicit expression of the rate of approximation.

Our approach is similar to [Devore and Ron, 2010]. The proof is decomposed onto several steps:

A.1.1 we first introduce a set $\mathcal{H}_\lambda \subset H^s(\mathbb{R}^p)$ and its restriction to $[0,1]^p$ and we derive some specificity of this space;

A.1.2 we prove the density of $\widetilde{\mathcal{V}}_n$ in $\mathcal{H}_\lambda$, considering functions defined on $\mathbb{R}^p$;

A.1.3 we deduce the approximation error of our procedure for functions belonging to $H^s([0,1]^p)$.

### C.2.1 The space $\mathcal{H}_\lambda$

For any $\lambda > 0$, we introduce the set

$$\mathcal{H}_\lambda := \left\{ g \in L^2(\mathbb{R}^p), \ \mathrm{supp}(\mathcal{F}g) \subset [-\lambda, \lambda]^p \right\}. \tag{1}$$

Note that $\mathcal{H}_\lambda \subset H^s(\mathbb{R}^p)$ for any $s > 0$. We also define

$$\mathcal{H}_\lambda|_{[0,1]^p} = \left\{ f \in L^2([0,1]^p) \text{ s.t. there exists } g \in \mathcal{H}_\lambda \text{ satisfying } g|_{[0,1]^p} = f \right\}.$$

We provide an integral representation of any function belonging to $\mathcal{H}_\lambda$.

**Lemma C.1.** *Let* $\lambda > 0$, $\boldsymbol{\sigma} \in (\mathbb{R}_+^*)^p$ *and* $g \in \mathcal{H}_\lambda$. *Set*

$$T_{\boldsymbol{\sigma}} g(\boldsymbol{\omega}) = \mathcal{F}^{-1} \left[ \frac{\mathcal{F}g(\boldsymbol{\omega})}{\mathcal{F}\phi_{\boldsymbol{\sigma}}^{[2]}(\boldsymbol{\omega})} \right].$$

*Then,*

$$g = \phi_{\boldsymbol{\sigma}}^{[2]} \star T_{\boldsymbol{\sigma}} g.$$

The proof of this lemma comes directly from the fact that for any function $g$ in $\mathcal{H}_\lambda$, one can define thanks to the assumption $\mathrm{supp}(\mathcal{F}\phi_{\boldsymbol{\sigma}}) = \mathbb{R}^p$

$$\frac{\mathcal{F}g(\boldsymbol{\omega})}{\mathcal{F}\phi_{\boldsymbol{\sigma}}^{[2]}(\boldsymbol{\omega})},$$

and that this function in a square integrable function. The remainder of the proof is left to the reader.

### C.2.2 Approximation of $\mathcal{H}_\lambda$ by $\widetilde{\mathcal{V}}_n$

For any $g \in \mathcal{H}_\lambda$, we introduce an approximation $\pi g \in \widetilde{\mathcal{V}}_n$, constructed through its coefficient $(A_k^{(M)})_{1 \leq k \leq K_n}$, for $M > 0$ fixed. This section ends with Proposition C.1, which controls the difference between $g$ and $\pi g$. In what follows, we denote for any $k \in \{1, \cdots, K_n\}$, $\boldsymbol{\xi}^{(k,M)}$ the center of $\mathcal{R}_k^{(n)} \cap [-M, M]^p$.

**Definition of an approximation of any $g \in \mathcal{H}_\lambda$ belonging to $\widetilde{\mathcal{V}}_n$**

**Lemma C.2.** *Let* $g \in \mathcal{H}_\lambda$. *Then*

(i) *For any* $k \in \{1, \cdots, K_n\}$, *the coefficients*

$$A_k^{(M)} := \int_{\mathbb{R}^p} \phi_{\boldsymbol{\sigma}}(\boldsymbol{\xi}^{(k,M)} - \mathbf{t}) T_{\boldsymbol{\sigma}} g(\mathbf{t}) d\mathbf{t} \tag{2}$$

*are well defined.*

*(ii) For any $k \in \{1, \cdots, K_n\}$,*

$$\max |A_k^{(M)}| \leq \int_{\mathbb{R}^p} \left| \frac{\mathcal{F}[g](\boldsymbol{\omega})}{\mathcal{F}[\phi_{\boldsymbol{\sigma}}](\boldsymbol{\omega})} \right| d\boldsymbol{\omega}.$$

*(iii) Define for all $\mathbf{x} \in \mathbb{R}^p$,*

$$\pi g(\mathbf{x}) = \sum_{k=1}^{K_n} A_k^{(M)} \widetilde{\Psi}(\mathbf{x}; \mathcal{R}_k^{(n)}, \boldsymbol{\sigma}).$$

*Then*

$$\sup_{\mathbf{x} \in \mathbb{R}^p} |\pi g(\mathbf{x})| \leq \int_{\mathbb{R}^p} \left| \frac{\mathcal{F}[g](\boldsymbol{\omega})}{\mathcal{F}[\phi_{\boldsymbol{\sigma}}](\boldsymbol{\omega})} \right| d\boldsymbol{\omega}.$$

**Proof:**

(i) Observe that formally

$$A_k^{(M)} = [\phi_{\boldsymbol{\sigma}} \star T_{\boldsymbol{\sigma}} g](\boldsymbol{\xi}^{(k,M)}) = \mathcal{F}^{-1}[\mathcal{F}[g]/\mathcal{F}[\phi_{\boldsymbol{\sigma}}]](\boldsymbol{\xi}^{(k,M)}).$$

Since $g \in \mathcal{H}_\lambda$, $\mathcal{F}[T_{\boldsymbol{\sigma}} g]$ belongs to $L^2(\mathbb{R}^p)$ and $\mathcal{F}[\phi_{\boldsymbol{\sigma}}] \in L^2(\mathbb{R}^p)$ then

$$\mathcal{F}[\phi_{\boldsymbol{\sigma}}]\mathcal{F}[T_{\boldsymbol{\sigma}} g] \in L^1(\mathbb{R}^p) .$$

In particular the function $\mathcal{F}^{-1}[\mathcal{F}[g]/\mathcal{F}[\phi_{\boldsymbol{\sigma}}]]$ is continuous on $\mathbb{R}^p$, which implies that the coefficients $A_k^{(M)}$ are well defined by (2), for all $k \in \{1, \ldots, K_n\}$.

(ii) Using the explicit expression of $A_k^{(M)}$ given in (i), one deduces that for any $k \in \{1, \cdots, K_n\}$,

$$|A_k^{(M)}| \quad \leq \quad \|\mathcal{F}^{-1}[\mathcal{F}[g]/\mathcal{F}[\phi_{\boldsymbol{\sigma}}]]\|_\infty \leq \|\mathcal{F}[g]/\mathcal{F}[\phi_{\boldsymbol{\sigma}}]\|_1 = \int_{\mathbb{R}^p} \left| \frac{\mathcal{F}[g](\boldsymbol{\omega})}{\mathcal{F}[\phi_{\boldsymbol{\sigma}}](\boldsymbol{\omega})} \right| d\boldsymbol{\omega}.$$

(iii) By definition of $\pi g$, one has for any $\mathbf{x} \in \mathbb{R}^p$,

$$|\pi g(\mathbf{x})| \leq \max_{k \in \{1, \ldots, K_n\}} |A_k^{(M)}| \cdot \left[ \sum_{k=1}^{K_n} \widetilde{\Psi}(\mathbf{x}; \mathcal{R}_k^{(n)}, \boldsymbol{\sigma}) \right].$$

Observe that, for all $\mathbf{x} \in \mathbb{R}^p$,

$$\sum_{k=1}^{K_n} \widetilde{\Psi}(\mathbf{x}; \mathcal{R}_k^{(n)}, \boldsymbol{\sigma}) = \int_{\mathbb{R}^p} \phi_{\boldsymbol{\sigma}}(\mathbf{u} - \mathbf{x}) d\mathbf{u} = 1,$$

by assumption on $\phi_{\boldsymbol{\sigma}}$. We now use the bound of the coefficients $A_k^{(M)}$ proved in (ii) to deduce (iii).

**Approximation of functions of $\mathcal{H}_\lambda$ with functions belonging to $\widetilde{\mathcal{V}}_n$**

**Proposition C.1.** *Let $M, \lambda > 0$ fixed and $g \in \mathcal{H}_\lambda$. Then, there exists some constant $C(\boldsymbol{\sigma}, p, \phi) > 0$, such that for any $s \in (1, 2)$ one has*

$$\forall \mathbf{x} \in [0, 1]^p, |g(\mathbf{x}) - \pi g(\mathbf{x})| \leq \frac{C(\boldsymbol{\sigma}, p, \phi)}{\inf_{|\boldsymbol{\omega}| \leq \lambda} |\mathcal{F}\phi_{\boldsymbol{\sigma}}(\boldsymbol{\omega})|} \|g\|_{H^s(\mathbb{R}^p)} \left[ \max_{k \in \{1, \ldots, K_n\}} \text{diam}(\mathcal{R}_k^{(n)} \cap [-M, M]^p) + M^{-r} \right].$$

(3)

**Proof:**
**Step 1: expression of the approximation error as a sum of integrals**
By Lemma C.1, for any $g \in \mathcal{H}_\lambda$, one has, for all $\mathbf{x} \in \mathbb{R}^p$, $g(\mathbf{x}) = \int_{\mathbb{R}^p} \phi_{\boldsymbol{\sigma}}^{[2]}(\mathbf{x} - \mathbf{t}) T_{\boldsymbol{\sigma}} g(\mathbf{t}) d\mathbf{t}$. We now use the
definition of $\phi_{\boldsymbol{\sigma}}$. As $(\mathcal{R}_k^{(n)})_{1 \le k \le K_n}$ is a partition of $\mathbb{R}^p$, one has for all $(\mathbf{x}, \mathbf{t}) \in (\mathbb{R}^p)^2$,

$$\phi_{\boldsymbol{\sigma}}^{[2]}(\mathbf{x} - \mathbf{t}) = \sum_{k=1}^{K_n} \int_{\mathcal{R}_k^{(n)}} \phi_{\boldsymbol{\sigma}}(\mathbf{x} - \mathbf{v}) \phi_{\boldsymbol{\sigma}}(\mathbf{v} - \mathbf{t}) d\mathbf{v}.$$

Thus,

$$g(\mathbf{x}) - \pi g(\mathbf{x}) = \sum_{k=1}^{K_n} \int_{\mathbf{v} \in \mathcal{R}_k^{(n)}} \phi_{\boldsymbol{\sigma}}(\mathbf{x} - \mathbf{v}) \left[ \int_{\mathbf{t} \in \mathbb{R}^p} \left[ \phi_{\boldsymbol{\sigma}}(\mathbf{v} - \mathbf{t}) - \phi_{\boldsymbol{\sigma}}(\boldsymbol{\xi}^{(k,M)} - \mathbf{t}) \right] T_{\boldsymbol{\sigma}} g(\mathbf{t}) d\mathbf{t} \right] d\mathbf{v}.$$

By the Mean Value Theorem, we get for any fixed $k \in \{1, \dots, K_n\}$,

$$\phi_{\boldsymbol{\sigma}}(\mathbf{v} - \mathbf{t}) - \phi_{\boldsymbol{\sigma}}(\boldsymbol{\xi}^{(k,M)} - \mathbf{t}) = \sum_{j=1}^{p} (v_j - \xi_j^{(k,M)}) \int_{z=0}^{1} z \times D_{v_j}[\phi_{\boldsymbol{\sigma}}] \left( z\mathbf{v} + (1-z)\boldsymbol{\xi}^{(k,M)} - \mathbf{t} \right) dz.$$

Since $\mathbb{1}_{\mathcal{R}_k^{(n)}} = \mathbb{1}_{\mathcal{R}_k^{(n)} \cap ([-M,M]^p)^c} + \mathbb{1}_{\mathcal{R}_k^{(n)} \cap [-M,M]^p}$, one has, for all $\mathbf{x} \in \mathbb{R}^p$,

$$g(\mathbf{x}) - \pi g(\mathbf{x}) = \sum_{k=1}^{K_n} \sum_{j=1}^{p} H_{k,j}^{(1)}(\mathbf{x}) + \sum_{k=1}^{K_n} \sum_{j=1}^{p} H_{k,j}^{(2)}(\mathbf{x}),$$

with for any $j \in \{1, \dots, p\}, k \in \{1, \dots, K_n\}$,

$$H_{k,j}^{(1)}(\mathbf{x}) = \int_{z=0}^{1} \int_{\mathbf{v} \in \mathbb{R}^p} z\phi_{\boldsymbol{\sigma}}(\mathbf{x} - \mathbf{v}) \times (v_j - \xi_j^{(k,M)}) \mathbb{1}_{\mathcal{R}_k^{(n)} \cap [-M,M]^p}(\mathbf{v}) [T_{\boldsymbol{\sigma}} g \star D_{v_j}[\phi_{\boldsymbol{\sigma}}]] \left( z\mathbf{v} + (1-z)\boldsymbol{\xi}^{(k,M)} \right) d\mathbf{v} dz, \tag{4}$$

and

$$H_{k,j}^{(2)}(\mathbf{x}) = \int_{z=0}^{1} \int_{\mathbf{v} \in \mathbb{R}^p} z\phi_{\boldsymbol{\sigma}}(\mathbf{x} - \mathbf{v}) \times (v_j - \xi_j^{(k,M)}) \mathbb{1}_{\mathcal{R}_k^{(n)} \cap ([-M,M]^p)^c}(\mathbf{v}) [T_{\boldsymbol{\sigma}} g \star D_{v_j}[\phi_{\boldsymbol{\sigma}}]](z\mathbf{v} + (1-z)\boldsymbol{\xi}^{(k,M)}) d\mathbf{v} dz. \tag{5}$$

**Step 2: bound of the sum $\sum_{k=1}^{K_n} \sum_{j=1}^{p} H_{k,j}^{(2)}$**

Eq. (5) implies that for all $\mathbf{x} \in [0,1]^p$, as $z \in [0,1]$,

$$|H_{k,j}^{(2)}(\mathbf{x})| \le \|T_{\boldsymbol{\sigma}} g \star D_{v_j}[\phi_{\boldsymbol{\sigma}}]\|_{L^2(\mathbb{R}^p)} \left[ \int_{\mathbf{v} \in \mathcal{R}_k^{(n)} \cap ([-M,M]^p)^c} |\phi_{\boldsymbol{\sigma}}(\mathbf{x} - \mathbf{v})|^2 \cdot |v_j - \xi_j^{(k,M)}|^2 d\mathbf{v} \right]^{1/2} \le \frac{C \|g\|_{H^s(\mathbb{R}^p)} M^{-r}}{\inf_{|\boldsymbol{\omega}| \le \lambda} |\mathcal{F}\phi_{\boldsymbol{\sigma}}(\boldsymbol{\omega})|}, \tag{6}$$

for some constant $C > 0$ depending only on $\phi, \boldsymbol{\sigma}$ and $p$. In the last display, we use Lemma C.5 to bound
$\|T_{\boldsymbol{\sigma}} g \star D_{v_j}[\phi_{\boldsymbol{\sigma}}]\|_{L^2(\mathbb{R}^p)}$ and Lemma C.3 to bound the integral involving $\phi_{\boldsymbol{\sigma}}$.

**Step 3: bound of the sum $\sum_{k=1}^{K_n} \sum_{j=1}^{p} H_{k,j}^{(1)}$**

The definition (4) of $H_{k,j}^{(1)}$ implies that for all $\mathbf{x} \in [0,1]^p$, as $z \in [0,1]$,

$$
\sum_{k=1}^{K_n} \sum_{j=1}^{p} H_{k,j}^{(1)} \leq \sum_{k=1}^{K_n} \sum_{j=1}^{p} \int_{z=0}^{1} \int_{\mathbf{v} \in \mathbb{R}^p} |\phi_{\boldsymbol{\sigma}}(\mathbf{x} - \mathbf{v})| \times \left| v_j - \xi_j^{(k,M)} \right|
$$
$$
\times \left| [T_{\boldsymbol{\sigma}} g \star D_{v_j}[\phi_{\boldsymbol{\sigma}}]](z\mathbf{v} + (1-z)\boldsymbol{\xi}^{(k,M)}) \right| \mathbb{1}_{\mathcal{R}_k^{(n)} \cap [-M,M]^p}(\mathbf{v}) d\mathbf{v} dz
$$
$$
\leq \left[ \max_{k \in \{1,\dots,K_n\}} \operatorname{diam}(\mathcal{R}_k^{(n)} \cap [-M,M]^p) \right] \times \sum_{k=1}^{K_n} \sum_{j=1}^{p} \int_{z=0}^{1} \int_{\mathbf{v} \in \mathbb{R}^p} |\phi_{\boldsymbol{\sigma}}(\mathbf{x} - \mathbf{v})|
$$
$$
\times \left| [T_{\boldsymbol{\sigma}} g \star D_{v_j}[\phi_{\boldsymbol{\sigma}}]](z\mathbf{v} + (1-z)\boldsymbol{\xi}^{(k,M)}) \right| \times \mathbb{1}_{\mathcal{R}_k^{(n)} \cap [-M,M]^p}(\mathbf{v}) d\mathbf{v} dz
$$
$$
\leq \left[ \max_{k \in \{1,\dots,K_n\}} \operatorname{diam}(\mathcal{R}_k^{(n)} \cap [-M,M]^p) \right] \times \sum_{k=1}^{K_n} \sum_{j=1}^{p} \int_{z=0}^{1} \left[ \int_{\mathbf{v} \in \mathbb{R}^p} |\phi_{\boldsymbol{\sigma}}(\mathbf{x} - \mathbf{v})| \right.
$$
$$
\times \left| T_{\boldsymbol{\sigma}} g \star D_{v_j}[\phi_{\boldsymbol{\sigma}}](z\mathbf{v} + (1-z)\boldsymbol{\xi}^{(k,M)}) - T_{\boldsymbol{\sigma}} g \star D_{v_j}[\phi_{\boldsymbol{\sigma}}](\mathbf{v}) \right| \mathbb{1}_{\mathcal{R}_k^{(n)} \cap [-M,M]^p}(\mathbf{v}) d\mathbf{v} \right] dz
$$
$$
+ \left[ \max_{k \in \{1,\dots,K_n\}} \operatorname{diam}(\mathcal{R}_k^{(n)} \cap [-M,M]^p) \right]
$$
$$
\times \sum_{k=1}^{K_n} \sum_{j=1}^{p} \left[ \int_{\mathbf{v} \in \mathbb{R}^p} |\phi_{\boldsymbol{\sigma}}(\mathbf{x} - \mathbf{v})| \left| [T_{\boldsymbol{\sigma}} g \star D_{v_j}[\phi_{\boldsymbol{\sigma}}]](\mathbf{v}) \right| \mathbb{1}_{\mathcal{R}_k^{(n)} \cap [-M,M]^p}(\mathbf{v}) d\mathbf{v} \right].
$$

We now bound each part. One first observes that

$$
\sum_{k=1}^{K_n} \sum_{j=1}^{p} \int_{\mathbf{v} \in \mathbb{R}^p} |\phi_{\boldsymbol{\sigma}}(\mathbf{x} - \mathbf{v})| \left| [T_{\boldsymbol{\sigma}} g \star D_{v_j}[\phi_{\boldsymbol{\sigma}}]](\mathbf{v}) \right| \mathbb{1}_{\mathcal{R}_k^{(n)} \cap [-M,M]^p}(\mathbf{v}) d\mathbf{v}
$$
$$
= \int_{\mathbf{v} \in \mathbb{R}^p} |\phi_{\boldsymbol{\sigma}}(\mathbf{x} - \mathbf{v})| \sum_{j=1}^{p} \left| [T_{\boldsymbol{\sigma}} g \star D_{v_j}[\phi_{\boldsymbol{\sigma}}]](\mathbf{v}) \right| d\mathbf{v} \leq \frac{C\|g\|_{H^s(\mathbb{R}^p)}}{\inf_{|\boldsymbol{\omega}| \leq \lambda} |\mathcal{F}\phi_{\boldsymbol{\sigma}}(\boldsymbol{\omega})|}.
$$

where in the last display we used Lemma C.5.
The other term is negligible. By the Cauchy Schwarz inequality, for any $z \in [0,1]$

$$\sum_{k=1}^{K_n}\sum_{j=1}^{p}\int_{\mathbf{v}\in\mathbb{R}^p}|\phi_{\boldsymbol{\sigma}}(\mathbf{x}-\mathbf{v})||T_{\boldsymbol{\sigma}}g\star D_{v_j}[\phi_{\boldsymbol{\sigma}}](z\mathbf{v}+(1-z)\boldsymbol{\xi}^{(k,M)})-T_{\boldsymbol{\sigma}}g\star D_{v_j}[\phi_{\boldsymbol{\sigma}}](\mathbf{v})|\mathbb{1}_{\mathcal{R}_k^{(n)}\cap[-M,M]^p}(\mathbf{v})d\mathbf{v}$$

$$=\sum_{k=1}^{K_n}\sum_{j=1}^{p}\int_{\mathbf{v}\in\mathcal{R}_k^{(n)}\cap[-M,M]^p}|\phi_{\boldsymbol{\sigma}}(\mathbf{x}-\mathbf{v})|\times|z\mathbf{v}+(1-z)\boldsymbol{\xi}^{(k,M)}-\mathbf{v}|^{s-1}$$

$$\times\frac{|T_{\boldsymbol{\sigma}}g\star D_{v_j}[\phi_{\boldsymbol{\sigma}}](z\mathbf{v}+(1-z)\boldsymbol{\xi}^{(k,M)})-T_{\boldsymbol{\sigma}}g\star D_{v_j}[\phi_{\boldsymbol{\sigma}}](\mathbf{v})|}{|z\mathbf{v}+(1-z)\boldsymbol{\xi}^{(k,M)}-\mathbf{v}|^{s-1}}d\mathbf{v}$$

$$\leq\;[\max_{k\in\{1,\ldots,K_n\}}\operatorname{diam}(\mathcal{R}_k^{(n)}\cap[-M,M]^p)]^{s-1}$$

$$\times\sum_{j=1}^{p}\int_{\mathbf{v}\in\mathbb{R}^p}|\phi_{\boldsymbol{\sigma}}(\mathbf{x}-\mathbf{v})|\times\frac{|T_{\boldsymbol{\sigma}}g\star D_{v_j}[\phi_{\boldsymbol{\sigma}}](z\mathbf{v}+(1-z)\boldsymbol{\xi}^{(k,M)})-T_{\boldsymbol{\sigma}}g\star D_{v_j}[\phi_{\boldsymbol{\sigma}}](\mathbf{v})|}{|z\mathbf{v}+(1-z)\boldsymbol{\xi}^{(k,M)}-\mathbf{v}|^{s-1}}d\mathbf{v}$$

$$\leq\;p[\max_{k\in\{1,\ldots,K_n\}}\operatorname{diam}(\mathcal{R}_k^{(n)}\cap[-M,M]^p)]^{s-1}\left[\int_{\mathbf{v}\in\mathbb{R}^p}|\phi_{\boldsymbol{\sigma}}(\mathbf{x}-\mathbf{v})|^2\,\mathbb{1}_{\mathcal{R}_k^{(n)}\cap[-M,M]^p}(\mathbf{v})d\mathbf{v}\right]^{1/2}$$

$$\times\left[\int_{\mathbf{v}\in\mathbb{R}^p}\frac{|T_{\boldsymbol{\sigma}}g\star D_{v_j}[\phi_{\boldsymbol{\sigma}}](z\mathbf{v}+(1-z)\boldsymbol{\xi}^{(k,M)})-T_{\boldsymbol{\sigma}}g\star D_{v_j}[\phi_{\boldsymbol{\sigma}}](\mathbf{v})|^2}{|z\mathbf{v}+(1-z)\boldsymbol{\xi}^{(k,M)}-\mathbf{v}|^{2(s-1)}}d\mathbf{v}\right]^{1/2}$$

$$\leq\;C[\max_{k\in\{1,\ldots,K_n\}}\operatorname{diam}(\mathcal{R}_k^{(n)}\cap[-M,M]^p)]^{s-1}\frac{\|\phi_{\boldsymbol{\sigma}}\|\cdot\|g\|_{H^s(\mathbb{R}^p)}}{\inf_{|\boldsymbol{\omega}|\leq\lambda}|\mathcal{F}\phi_{\boldsymbol{\sigma}}(\boldsymbol{\omega})|}.$$

where the last inequality comes from Lemma C.5 and Lemma C.3, with $C$ is a constant depending only on $\phi,\boldsymbol{\sigma}$ and $p$. As $\phi_{\boldsymbol{\sigma}}$ is a probability distribution function, we then deduce the following bound for (4):

$$\sum_{k=1}^{K_n}\sum_{j=1}^{p}H_{k,j}^{(1)}\leq\frac{C(\boldsymbol{\sigma},p,\phi)\|g\|_{H^s(\mathbb{R}^p)}}{\inf_{|\boldsymbol{\omega}|\leq\lambda}|\mathcal{F}\phi_{\boldsymbol{\sigma}}(\boldsymbol{\omega})|}\cdot[\max_{k\in\{1,\ldots,K_n\}}\operatorname{diam}(\mathcal{R}_k^{(n)}\cap[-M,M]^p)].\tag{7}$$

The respective bounds (6) and (7) end the proof of Eq. (3).

**Extension to the density of $\mathcal{V}_n$ in $H^s([0,1]^p)$**
We are now given $f\in H^s([0,1]^p)$. As we are considering

$$\|f\|_{H^s([0,1]^P)}=\inf\{\|g\|_{H^s(\mathbb{R}^p)},\,g\in H^s(\mathbb{R}^p),\;\text{s.t.}\;f=g|_{[0,1]^p}\},$$

we consider $g$ an extension of $f$ on $\mathbb{R}^p$ belonging to $H^s(\mathbb{R}^p)$. Let $(\lambda_n)$ be a non decreasing sequence converging to $+\infty$. As in [Schaback, 1995], we consider $g_{\lambda_n}\in\mathcal{H}_{\lambda_n}$ the approximation of $g$ defined in Lemma C.6. We set

$$f_{\lambda_n}=g_{\lambda_n}|_{[0,1]^p}\text{ and }h_{\lambda_n}=(\pi g_{\lambda_n})|_{[0,1]^p}.$$

By definition $f_{\lambda_n}\in\mathcal{H}_{\lambda_n}|_{[0,1]^p}$. Moreover,

$$\|h_{\lambda_n}\|_{L^\infty([0,1]^P)}=\|\pi g_{\lambda_n}\|_{L^\infty([0,1]^P)}\leq\frac{\lambda_n^{p/2}\|g\|_{H^s(\mathbb{R}^p)}}{\inf_{|\boldsymbol{\omega}|\leq\lambda_n}|\mathcal{F}\phi_{\boldsymbol{\sigma}}(\boldsymbol{\omega})|},$$

where the last inequality comes from Lemma C.6[(ii)]. If we assume that

$$\beta_n>\frac{\|f\|_{H^s([0,1]^P)}\cdot\lambda_n^{p/2}}{\inf_{|\boldsymbol{\omega}|\leq\lambda_n}|\mathcal{F}[\phi_{\boldsymbol{\sigma}}](\boldsymbol{\omega})|},\tag{8}$$

and consider an extension $g$ such that $\|g\|_{H^s(\mathbb{R}^p)}$ norm is sufficiently close from $\|f\|_{H^s([0,1]^p)}$, one has $h_{\lambda_n} \in \mathcal{T}_{\beta_n} \mathcal{V}_n$. Hence, one has

$$
\inf_{h \in \mathcal{T}_{\beta_n} \mathcal{V}_n} \|f - h\|_{L^2([0,1]^p)} \leq \|f - h_{\lambda_n}\|_{L^2([0,1]^p)} \leq \|f - f_{\lambda_n}\|_{L^2([0,1]^p)} + \|f_{\lambda_n} - h_{\lambda_n}(x)\|_{L^2([0,1]^p)}
$$

$$
= \|g - g_{\lambda_n}\|_{L^2([0,1]^p)} + \|g_{\lambda_n} - \pi g_{\lambda_n}\|_{L^2([0,1]^p)}
$$

$$
\leq \frac{1}{(1 + |\lambda_n|^2)^{s/2}} \|g\|_{H^s(\mathbb{R}^p)}
$$

$$
+ \frac{C(\boldsymbol{\sigma}, p, \phi) \|g_{\lambda_n}\|_{H^s(\mathbb{R}^p)}}{\inf_{|\boldsymbol{\omega}| \leq \lambda_n} |\mathcal{F}\phi_{\boldsymbol{\sigma}}(\boldsymbol{\omega})|} \left( [\max_{k \in \{1,\dots,K_n\}} \operatorname{diam}(\mathcal{R}_k^{(n)} \cap [-M, M]^p)] + M^{-r} \right)
$$

$$
\leq C(\boldsymbol{\sigma}, p, \phi) \|g\|_{H^s(\mathbb{R}^p)} \left[ \frac{1}{(1 + |\lambda_n|^2)^{s/2}} + \beta_n [\max_{k \in \{1,\dots,K_n\}} \operatorname{diam}(\mathcal{R}_k^{(n)} \cap [-M, M]^p) + M^{-r}] \right]
$$

where in the last inequality we use Assumption (8) on $(\beta_n)$, Lemma C.6 and Proposition C.1. Since

$$
\|f\|_{H^s([0,1]^p)} = \inf\{\|g\|_{H^s(\mathbb{R}^p)}, \, g \in H^s(\mathbb{R}^p), \text{ s.t. } f = g|_{[0,1]^p}\};
$$

it leads to the following proposition:

**Proposition C.2.** *Let $s \in (1,2)$, $M > 0$ and $f \in H^s([0,1]^p)$. Let $(\beta_n)$ be a non decreasing sequence converging to $\infty$. Consider a non decreasing sequence $(\lambda_n)$ such that for $n$ sufficiently large,*

$$
\beta_n > \frac{\|f\|_{H^s([0,1]^p)} \cdot \lambda_n^{p/2}}{\inf_{|\boldsymbol{\omega}| \leq \lambda_n} |\mathcal{F}[\phi_{\boldsymbol{\sigma}}](\boldsymbol{\omega})|}. \tag{9}
$$

*Then, for some $C(\boldsymbol{\sigma}, p, \phi) > 0$, one has*

$$
\inf_{h \in \mathcal{T}_{\beta_n} \mathcal{V}_n} \|f - h\|_{L^2([0,1]^p)} \leq C(\boldsymbol{\sigma}, p, \phi) \|f\|_{H^s(\mathbb{R}^p)} \left[ \frac{1}{(1 + |\lambda_n|^2)^{s/2}} + \beta_n [\max_{k \in \{1,\dots,K_n\}} \operatorname{diam}(\mathcal{R}_k^{(n)} \cap [-M, M]^p) + M^{-r}] \right].
$$

### C.2.3 Extension to Proposition 1 in the main paper

Setting $f = \mathbb{E}(Y|\mathbf{X} = \cdot)$, one can easily deduce the following proposition.

**Proposition C.3** (Proposition 1 in the main paper). *Assume that for some $s \in (1,2)$, a.s. $\mathbb{E}(Y|\mathbf{X} = \cdot) \in H^s([0,1]^p)$. Let $M > 0$ and $(\beta_n)$ be a non decreasing sequence converging to $\infty$. Consider a non decreasing sequence $(\lambda_n)$ such that for $n$ sufficiently large, Condition (9) holds. Then, one has for an absolute constant $C$,*

$$
\inf_{g \in \mathcal{T}_{\beta_n} \mathcal{V}_n} \|\mathbb{E}(Y|\mathbf{X} = \cdot) - g\|_{L^2([0,1]^p)}^2 \leq C \|\mathbb{E}(Y|\mathbf{X} = \cdot)\|_{H^s([0,1]^p)}^2 \, a_n, \tag{10}
$$

*with*

$$
a_n = \left( \frac{1}{(1 + |\lambda_n|^2)^{s/4}} + \beta_n \mathbb{E}\left[ \left( \max_{k=1,\dots,K_n} \operatorname{diam}(\mathcal{R}_k^{(n)} \cap [-M, M]^p) \right) \right] + \beta_n M^{-r} \right)^2.
$$

### C.2.4 Technical lemmas

**Lemma C.3.** *Let $(\sigma_1, \cdots, \sigma_p) \in (\mathbb{R}_+^*)^p$. Assume that*

$$
\sup_{\mathbf{v} \in \mathbb{R}^p} |\mathbf{v}|^{1+r+p/2} |\phi_{\boldsymbol{\sigma}}(\mathbf{v})| < +\infty. \tag{11}
$$

*Then there exists a constant $C > 0$ depending only on $\phi, \boldsymbol{\sigma}$ and $p$ such that for any $(\mathbf{x}, \boldsymbol{\xi}) \in ([0,1]^p)^2$ and $M \geq 2$.*

$$
\int_{\mathbf{v} \in ([-M,M]^p)^c} |\phi_{\boldsymbol{\sigma}}(\mathbf{x} - \mathbf{v})|^2 |v_j - \xi_j|^2 d\mathbf{v} \leq C M^{-2r} \tag{12}
$$

**Proof:** Since $M \geq 2$, for each $j$ and any $(\mathbf{x}, \boldsymbol{\xi}) \in ([0,1]^p)^2$ and $\mathbf{v} \in ([-M, M]^p)^c$, observe that for $M > 0$ sufficiently large

$$([-M, M]^p)^c - x \subset ([-M/2, M/2]^p)^c$$

and

$$|v_j - \xi_j| \leq C|v_j - x_j| \, .$$

where the constant $C > 0$. Hence, combining the change of variable $\mathbf{v} \leftarrow \mathbf{x} - \mathbf{v}$ and the previous remarks, one deduces that

$$
\begin{aligned}
\int_{\mathbf{v} \in ([-M,M]^p)^c} |\phi_{\boldsymbol{\sigma}}(\mathbf{x} - \mathbf{v})|^2 \, |v_j - \xi_j|^2 d\mathbf{v} &\leq C \int_{\mathbf{v} \in ([-M,M]^p)^c} |\phi_{\boldsymbol{\sigma}}(\mathbf{x} - \mathbf{v})|^2 \, |v_j - x_j|^2 d\mathbf{v} \\
&\leq C \int_{\mathbf{v} \in ([-M/2,M/2]^p)^c} |\phi_{\boldsymbol{\sigma}}(\mathbf{v})|^2 \, |v_j|^2 d\mathbf{v} \\
&\leq C \int_{\mathbf{v} \in ([-M/2,M/2]^p)^c} |\mathbf{v}|^{-2r-p} d\mathbf{v};
\end{aligned}
$$

where the last display comes from Assumption (12) on $\phi_{\boldsymbol{\sigma}}$. It ends the proof.

We also give an approximation result of any function belonging to $H^s(\mathbf{R}^p)$ by a function in $\mathcal{H}_\lambda$.

**Lemma C.4.** *Let $\lambda > 0$, $s > 1$ and $f \in H^s(\mathbb{R}^p)$. One has, for any $\alpha \in (0, s-1)$,*

$$\left\| \mathbf{x} \mapsto f(\mathbf{x}) - (2\pi)^{-p} \int_{|\boldsymbol{\omega}| \leq \lambda} \mathcal{F}(f)(\boldsymbol{\omega}) e^{i\boldsymbol{\omega} \cdot \mathbf{x}} d\omega \right\|_\infty \leq \frac{C(\alpha)(2\pi)^{-p}}{(1 + |\lambda|^2)^{(s-1-\alpha)/2}} \|f\|_{H^s(\mathbb{R}^p)}.$$

**Proof:**
We follow the same line as [Schaback, 1995]. One has, for all $\mathbf{x} \in \mathbb{R}^p$,

$$\left| f(\mathbf{x}) - (2\pi)^{-p} \int_{|\boldsymbol{\omega}| \leq \lambda} \mathcal{F}(f)(\boldsymbol{\omega}) e^{i\boldsymbol{\omega} \cdot \mathbf{x}} d\boldsymbol{\omega} \right| \leq (2\pi)^{-p} \int_{|\boldsymbol{\omega}| \geq \lambda} |\mathcal{F}(f)(\boldsymbol{\omega})| d\boldsymbol{\omega}$$

$$\leq \frac{(2\pi)^{-p}}{(1 + |\lambda|^2)^{(s-1-\alpha)/2}} \int_{|\boldsymbol{\omega}| \geq \lambda} \frac{(1 + |\boldsymbol{\omega}|^2)^{s/2}}{(1 + |\boldsymbol{\omega}|^2)^{(1+\alpha)/2}} |\mathcal{F}(f)(\boldsymbol{\omega})| d\boldsymbol{\omega} \leq \frac{C(\alpha)(2\pi)^{-p}}{(1 + |\lambda|^2)^{(s-1-\alpha)/2}} \|f\|_{H^s(\mathbb{R}^p)},$$

using Cauchy-Schwarz in the last inequality.

**Lemma C.5.** *Let $\lambda > 0$ and $g \in \mathcal{H}_\lambda$. Then, for all $j \in \{1, \ldots, p\}$, $s \in (1, 2)$, one has*

$$\|T_{\boldsymbol{\sigma}} g \star D_{v_j}[\phi_{\boldsymbol{\sigma}}]\|_{L^2(\mathbb{R}^p)} + \sup_{|\mathbf{h}| \leq 1} \int_{\mathbf{v} \in \mathbb{R}^p} \frac{|T_{\boldsymbol{\sigma}} g \star D_{v_j}[\phi_{\boldsymbol{\sigma}}](\mathbf{v} + \mathbf{h}) - T_{\boldsymbol{\sigma}} g \star D_{v_j}[\phi_{\boldsymbol{\sigma}}](\mathbf{v})|^2}{|\mathbf{h}|^{2(s-1)}} d\mathbf{v} \leq \frac{C \|g\|_{H^s(\mathbb{R}^p)}}{\inf_{|\boldsymbol{\omega}| \leq \lambda} |\mathcal{F}\phi_{\boldsymbol{\sigma}}(\boldsymbol{\omega})|},$$

*where $C$ is a positive constant depending only on $\phi, \boldsymbol{\sigma}$ and $p$.*

**Proof:**
One has, for $\boldsymbol{\omega} \in \mathbb{R}^p$ and any $s > 1$,

$$\|T_{\boldsymbol{\sigma}} g \star D_{v_j}[\phi_{\boldsymbol{\sigma}}]\|_{H^{s-1}(\mathbb{R}^p)} = \left( \int (1 + |\boldsymbol{\omega}|^2)^{s-1} |\mathcal{F}[T_{\boldsymbol{\sigma}} g \star D_{v_j}[\phi_{\boldsymbol{\sigma}}]](\boldsymbol{\omega})|^2 d\boldsymbol{\omega} \right)^{1/2}.$$

However, for all $\boldsymbol{\omega} \in \mathbb{R}^p$,

$$
\begin{aligned}
(1 + |\boldsymbol{\omega}|^2)^{s-1} |\mathcal{F}[T_{\boldsymbol{\sigma}} g \star D_{v_j}[\phi_{\boldsymbol{\sigma}}]](\boldsymbol{\omega})|^2 &= (1 + |\boldsymbol{\omega}|^2)^{s-1} |\mathcal{F}[T_{\boldsymbol{\sigma}} g](\boldsymbol{\omega})|^2 |\mathcal{F}[D_{v_j}[\phi_{\boldsymbol{\sigma}}]](\boldsymbol{\omega})|^2 \\
&\leq C(1 + |\boldsymbol{\omega}|^2)^s |\mathcal{F}[T_{\boldsymbol{\sigma}} g](\boldsymbol{\omega})|^2 |\mathcal{F}[\phi_{\boldsymbol{\sigma}}](\boldsymbol{\omega})|^2 \\
&\leq C(1 + |\boldsymbol{\omega}|^2)^s |\mathcal{F}[g](\boldsymbol{\omega})|^2 / |\mathcal{F}[\phi_{\boldsymbol{\sigma}}](\boldsymbol{\omega})|^2.
\end{aligned}
$$

where $C$ is a positive constant depending only on $\boldsymbol{\sigma}$. In the last display we use the explicit expressions of $T_{\boldsymbol{\sigma}}g$ and that of the Fourier transform of $D_{v_j}[\phi_{\boldsymbol{\sigma}}]$. Since $\phi_{\boldsymbol{\sigma}} \in L^1(\mathbb{R}^p)$, $\mathcal{F}[\phi_{\boldsymbol{\sigma}}] \in L^\infty(\mathbb{R}^p)$. In addition, the band limited assumption $g \in \mathcal{H}_\lambda$ implies

$$|\mathcal{F}[T_{\boldsymbol{\sigma}}g](\boldsymbol{\omega})|^2 \leq \frac{C|\mathcal{F}[g](\boldsymbol{\omega})|^2}{\inf_{|\boldsymbol{\omega}|\leq\lambda}|\mathcal{F}\phi_{\boldsymbol{\sigma}}(\boldsymbol{\omega})|^2}$$

Hence we get the result.

**Lemma C.6.** *Let* $g \in H^s(\mathbb{R}^p)$ *and* $\lambda > 0$. *Set for all* $\mathbf{x} \in \mathbb{R}^p$, $g_\lambda(\mathbf{x}) = (2\pi)^{-p} \int_{|\boldsymbol{\omega}|\leq\lambda} \mathcal{F}g(\boldsymbol{\omega})e^{i\boldsymbol{\omega}\cdot\mathbf{x}}d\boldsymbol{\omega}$. *Then for some* $C > 0$ *depensing only on* $s$, *one has*

(i) $\|g_\lambda\|^2_{L^2(\mathbb{R}^p)} \leq C\|g\|^2_{H^s(\mathbb{R}^p)}$;

(ii) $\|\pi g_\lambda\|_{L^\infty([0,1]^p)} \leq \frac{\|g\|_{H^s(\mathbb{R}^p)}\cdot\lambda^{p/2}}{\inf_{|\boldsymbol{\omega}|\leq\lambda}|\mathcal{F}[\phi_{\boldsymbol{\sigma}}](\boldsymbol{\omega})|}$;

(iii) $\|g_\lambda - g\|^2_{L^2(\mathbb{R}^p)} \leq \frac{C}{(1+|\lambda|^2)^s} \cdot \|g\|^2_{H^s(\mathbb{R}^p)}$.

**Proof:**

(i) Using Parseval equality one has

$$
\begin{aligned}
\|g_\lambda\|^2_{L^2(\mathbb{R}^p)} &= \|\mathcal{F}g_\lambda\|^2_{L^2(\mathbb{R}^p)} = \int_{|\boldsymbol{\omega}|\leq\lambda}|\mathcal{F}g(\boldsymbol{\omega})|^2 d\boldsymbol{\omega} \\
&\leq \int_{\mathbb{R}^p}|\mathcal{F}g(\boldsymbol{\omega})|^2 d\boldsymbol{\omega} \\
&\leq \|g\|^2_{L^2(\mathbb{R}^p)}.
\end{aligned}
$$

By usual Sobolev embedding, one has $\|g\|^2_{L^2(\mathbb{R}^p)} \leq C\|g\|^2_{H^s(\mathbb{R}^p)}$ and deduce (i).

(ii) Observe that

$$
\begin{aligned}
\|\pi g_\lambda\|_{L^\infty([0,1]^p)} &\leq \int_{\mathbb{R}^p}\frac{|\mathcal{F}[g_\lambda](\boldsymbol{\omega})|}{|\mathcal{F}[\phi_{\boldsymbol{\sigma}}](\boldsymbol{\omega})|}d\boldsymbol{\omega} \\
&\leq \left(\int_{\mathbb{R}^p}\frac{\mathbb{1}_{|\boldsymbol{\omega}|\leq\lambda}}{|\mathcal{F}[\phi_{\boldsymbol{\sigma}}](\boldsymbol{\omega})|^2}d\boldsymbol{\omega}\right)^{1/2} \times \left(\int_{\mathbb{R}^p}|\mathcal{F}[g_\lambda](\boldsymbol{\omega})|^2 d\boldsymbol{\omega}\right)^{1/2} \\
&\leq \frac{\lambda^{p/2}}{\inf_{|\boldsymbol{\omega}|\leq\lambda}|\mathcal{F}[\phi_{\boldsymbol{\sigma}}](\boldsymbol{\omega})|}\cdot\|g_\lambda\|_{L^2(\mathbb{R}^p)} \\
&\leq C\frac{\lambda^{p/2}\cdot\|g\|_{H^s(\mathbb{R}^p)}}{\inf_{|\boldsymbol{\omega}|\leq\lambda}|\mathcal{F}[\phi_{\boldsymbol{\sigma}}](\boldsymbol{\omega})|};
\end{aligned}
$$

where the last display comes from (i).

(iii) In the same way

$$
\begin{aligned}
\|g_\lambda - g\|^2_{L^2(\mathbb{R}^p)} &= \|\mathcal{F}g_\lambda - \mathcal{F}g\|^2_{L^2(\mathbb{R}^p)} = \int_{|\boldsymbol{\omega}|\geq\lambda}|\mathcal{F}g(\boldsymbol{\omega})|^2 d\boldsymbol{\omega} \\
&= \int_{|\boldsymbol{\omega}|\geq\lambda}\frac{1}{(1+|\boldsymbol{\omega}|^2)^s}\cdot(1+|\boldsymbol{\omega}|^2)^s|\mathcal{F}g(\boldsymbol{\omega})|^2 d\boldsymbol{\omega} \\
&\leq \frac{1}{(1+|\lambda|^2)^s}\cdot\|g\|^2_{H^s(\mathbb{R}^p)}.
\end{aligned}
$$

## C.3 Proof of Proposition 2 of the main paper: estimation error

This part is based on [Györfi et al., 2002].

We define $Y_{\beta_n} = \mathcal{T}_{\beta_n} Y$, $Y_{\beta_n}^{(i)} = \mathcal{T}_{\beta_n} Y^{(i)}$ with the truncated operator $\mathcal{T}$ introduced before. We are also given $U = (\mathbf{X}, Y)$, and we consider $U^{(1)} = (\mathbf{X}^{(1)}, Y^{(1)}), \ldots, U^{(n)} = (\mathbf{X}^{(n)}, Y^{(n)})$ a sequence of iid copies of $U = (\mathbf{X}, Y)$, and $(u^{(1)}, \ldots, u^{(n)})$ the corresponding realizations of these random variables.

**Proposition C.4** (Proposition 2 of the main paper). *Let $(\beta_n)$ be a sequence of non negative numbers such that $\lim_{n \to +\infty} \beta_n = +\infty$. For any $n \in \mathbb{N}$, for $\eta > 0$,*

$$\mathbb{E}\left[ \sup_{g \in \mathcal{T}_{\beta_n} \mathcal{V}_n} \left| \frac{1}{n} \sum_{i=1}^n |g(\mathbf{X}^{(i)}) - \mathcal{T}_{\beta_n} Y^{(i)}|^2 - \mathbb{E}[|g(\mathbf{X}) - \mathcal{T}_{\beta_n} Y|^2] \right| \right] \le b_n, \tag{13}$$

*with $b_n = \eta + 64\beta_n^2 \exp[-c_n]$,*

$$c_n = \frac{n}{\beta_n^4} \left\{ \frac{\eta^2}{2048} - \frac{\beta_n^4 (K_n - 1) \log(pn)}{n} - \frac{2\beta_n^4 K_n}{n} \log\left( \frac{C'\beta_n^2}{\eta} \right) \right\};$$

*and $C' > 0$ is a universal constant.*

**Proof:**

We recall here the main steps and the specific extension we need in our setting.

Let
$$\mathcal{A}_n = \{ a : \mathbb{R}^p \times \mathbb{R} \to \mathbb{R} : \exists g \in \mathcal{T}_{\beta_n} \mathcal{V}_n \text{ s.t. } a(\mathbf{x}, y) = |g(\mathbf{x}) - \mathcal{T}_{\beta_n}(y)|^2 \}.$$

First, remark that for all $a \in \mathcal{A}_n$, $0 \le a(\mathbf{x}, y) \le 4\beta_n^2$.

Let $\eta > 0$. For any set $\mathcal{G}$ of functions defined on $[0,1]^p \times \mathbb{R}$, we denote by $\mathcal{N}_1(\eta, \mathcal{G}, (u^{(1)}, \ldots, u^{(n)}))$ an $L^1([0,1]^p \times \mathbb{R})$ $\eta$-cover of $\mathcal{G}$ on $(u^{(1)}, \ldots, u^{(n)})$, corresponding to the minimum number of functions needed to approximate any function of $\mathcal{G}$, with respect to the $L^1([0,1]^p \times \mathbb{R})$-norm on observed points $(u^{(1)}, \ldots, u^{(n)})$ through the empirical measure $\nu_n$, $\|f\|_{L_1(\nu_n)} = \sum_{i=1}^n |f(u^{(i)})|/n$, up to $\eta$. From [Györfi et al., 2002, Theorem 9.1], we get

$$\mathbb{P}\left( \sup_{g \in \mathcal{T}_{\beta_n} \mathcal{V}_n} \left| \frac{1}{n} \sum_{i=1}^n |g(\mathbf{X}^{(i)}) - Y_{\beta_n}^{(i)}|^2 - \mathbb{E}[|g(\mathbf{X}) - Y_{\beta_n}|^2] \right| > \eta \right)$$

$$\le 8\mathcal{N}_1(\eta/8, \mathcal{A}_n, (U^{(1)}, \ldots, U^{(n)})) \exp\left( -\frac{n\eta^2}{128(4\beta_n^2)^2} \right).$$

We now bound $\mathcal{N}_1(\eta/8, \mathcal{A}_n, (U^{(1)}, \ldots, U^{(n)}))$.

Using [Györfi et al., 2002, Problem 10.4] for the first inequality and [Györfi et al., 2002, Lemma 13.1, Theorem 9.4] for the second inequality,

$$\mathcal{N}_1\left( \frac{\eta}{8}, \mathcal{A}_n, (U^{(1)}, \ldots, U^{(n)}) \right) \le \mathcal{N}_1\left( \frac{\eta}{32\beta_n}, T_{\beta_n} \mathcal{V}_n, (\mathbf{X}^{(1)}, \ldots, \mathbf{X}^{(n)}) \right) \le (pn)^{K_n - 1} \left\{ \left( \frac{C'\beta_n^2}{\eta} \right)^2 \right\}^{K_n},$$

where $C'$ is a universal constant. It comes from several combinatorial considerations: there are $K_n$ nodes in the tree, the estimator constructed in each region is a constant, and the partition is constructed by choosing a direction (among $p$) and a split value (among $n$) at each step (til $K_n$ regions, so $K_n - 1$ steps). Then, we get

$$\mathbb{P}\left( \sup_{g \in \mathcal{T}_{\beta_n} \mathcal{V}_n} \left| \frac{1}{n} \sum_{i=1}^n |g(\mathbf{X}^{(i)}) - Y_{\beta_n}^{(i)}|^2 - \mathbb{E}[|g(\mathbf{X}) - Y_{\beta_n}|^2] \right| > \eta \right)$$

$$\le 8(pn)^{K_n - 1} \left( \frac{C'\beta_n^2}{\eta} \right)^{2K_n} \exp\left( -\frac{n\eta^2}{2048\beta_n^4} \right)$$

$$\le 8 \exp\left[ -\frac{n}{\beta_n^4} \left\{ \frac{\eta^2}{2048} - \frac{\beta_n^4 (K_n - 1) \log(pn)}{n} - \frac{2\beta_n^4 K_n}{n} \log\left( \frac{C'\beta_n^2}{\eta} \right) \right\} \right] := b_n^{(1)}.$$

Observe that since $g \in \mathcal{T}_{\beta_n} \mathcal{V}_n$, one has $\|g\|_\infty \leq \beta_n$. Similarly, $|Y_{\beta_n}| \leq \beta_n$. Hence

$$\sup_{g \in \mathcal{T}_{\beta_n} \mathcal{V}_n} \left| \frac{1}{n} \sum_{i=1}^{n} |g(\mathbf{X}^{(i)}) - Y_{\beta_n}^{(i)}|^2 - \mathbb{E}[|g(\mathbf{X}) - Y_{\beta_n}|^2] \right| \leq 8\beta_n^2,$$

so we get the following:

$$\mathbb{E} \left( \sup_{g \in \mathcal{T}_{\beta_n} \mathcal{V}_n} \left| \frac{1}{n} \sum_{i=1}^{n} |g(\mathbf{X}^{(i)}) - Y_{\beta_n}^{(i)}|^2 - \mathbb{E}[|g(\mathbf{X}) - Y_{\beta_n}|^2] \right| \right)$$

$$\leq \mathbb{E} \left( \sup_{g \in \mathcal{T}_{\beta_n} \mathcal{V}_n} \left| \frac{1}{n} \sum_{i=1}^{n} |g(\mathbf{X}^{(i)}) - Y_{\beta_n}^{(i)}|^2 - \mathbb{E}[|g(\mathbf{X}) - Y_{\beta_n}|^2] \right| \mathbb{1}_{\sup_{g \in \mathcal{T}_{\beta_n} \mathcal{V}_n} \left| \frac{1}{n} \sum_{i=1}^{n} |g(\mathbf{X}^{(i)}) - Y_{\beta_n}^{(i)}|^2 - \mathbb{E}[|g(\mathbf{X}) - Y_{\beta_n}|^2] \right| \leq \eta} \right)$$

$$+ \mathbb{E} \left( \sup_{g \in \mathcal{T}_{\beta_n} \mathcal{V}_n} \left| \frac{1}{n} \sum_{i=1}^{n} |g(\mathbf{X}^{(i)}) - Y_{\beta_n}^{(i)}|^2 - \mathbb{E}[|g(\mathbf{X}) - Y_{\beta_n}|^2] \right| \mathbb{1}_{\sup_{g \in \mathcal{T}_{\beta_n} \mathcal{V}_n} \left| \frac{1}{n} \sum_{i=1}^{n} |g(\mathbf{X}^{(i)}) - Y_{\beta_n}^{(i)}|^2 - \mathbb{E}[|g(\mathbf{X}) - Y_{\beta_n}|^2] \right| \geq \eta} \right)$$

$$\leq \eta + 8\beta_n^2 b_n^{(1)} := b_n.$$

It ends the proof.

## C.4 Proof of Theorem 1 of the main paper

### C.4.1 Consequence of Propositions C.3 and C.4

We first deduce from Proposition C.3 and C.4 the following result

**Theorem C.1.** *With the assumptions and notations of Proposition and Proposition C.4*

$$\mathbb{E}_{\mathcal{D}_n} \left[ \mathbb{E}[|\widetilde{T}_s^{(n)}(\mathbf{X}) - Y|^2 | \mathcal{D}_n]^{1/2} - \mathbb{E}[|\mathbb{E}(Y|\mathbf{X}) - Y|^2]^{1/2} \right]$$

$$\leq 4a_n + 12b_n + 12\mathbb{E}[|Y - Y_{\beta_n}|^2]$$

$$+ \frac{12}{n} \sum_{i=1}^{n} \mathbb{E}[|Y^{(i)} - Y_{\beta_n}^{(i)}|^2],$$

*with $Y_{\beta_n} = \mathcal{T}_{\beta_n} Y$ and $Y_{\beta_n}^{(i)} = \mathcal{T}_{\beta_n} Y^{(i)}$.*

**Proof**
Our proof is based on a modified version of [Györfi et al., 2002, Theorem 10.2b]. Recall that $\widetilde{T}_s^{(n)} = \mathcal{T}_{\beta_n} \hat{T}_s^{(n)}$ is the truncated estimator.

$$\mathbb{E} \left[ \int_{[0,1]^d} |\widetilde{T}_s^{(n)}(\mathbf{x}) - \mathbb{E}(Y|\mathbf{X} = \mathbf{x})|^2 d\mathbb{P}(\mathbf{x}) \right] = \left[ \mathbb{E}[|\widetilde{T}_s^{(n)}(\mathbf{X}) - Y|^2 | \mathcal{D}_n]^{1/2} - \mathbb{E}[|\mathbb{E}(Y|\mathbf{X}) - Y|^2]^{1/2} \right]^2$$

$$+ 2\mathbb{E}[|\mathbb{E}(Y|\mathbf{X}) - Y|^2]^{1/2} \cdot \left[ \mathbb{E}[|\widetilde{T}_s^{(n)}(\mathbf{X}) - Y|^2 | \mathcal{D}_n]^{1/2} - \mathbb{E}[|\mathbb{E}(Y|\mathbf{X}) - Y|^2]^{1/2} \right].$$

To get the conclusion of Theorem 1, we then have to bound

$$\mathbb{E}_{\mathcal{D}_n} \left[ \mathbb{E}[|\widetilde{T}_s^{(n)}(\mathbf{X}) - Y|^2 | \mathcal{D}_n]^{1/2} - \mathbb{E}[|\mathbb{E}(Y|\mathbf{X}) - Y|^2]^{1/2} \right] .$$

We use the following error decomposition used in the proof of [Györfi et al., 2002, Theorem 10.2b] that we

recall

$$\mathbb{E}_{\mathcal{D}_n}\left[\mathbb{E}[|\widetilde{T}_s^{(n)}(\mathbf{X}) - Y|^2|\mathcal{D}_n]^{1/2} - \mathbb{E}[|\mathbb{E}(Y|\mathbf{X}) - Y|^2]^{1/2}\right]$$

$$\leq \quad 2\mathbb{E}[\inf_{g\in\mathcal{T}_{\beta_n}\mathcal{V}_n}\int_{[0,1]^p}|g(\mathbf{x}) - \mathbb{E}(Y|\mathbf{X} = \mathbf{x})|^2 d\mathbf{x}] + 12\mathbb{E}[|Y - Y_{\beta_n}|^2] + \frac{12}{n}\sum_{i=1}^n \mathbb{E}[|Y^{(i)} - Y_{\beta_n}^{(i)}|^2]$$

$$+12\mathbb{E}\left[\sup_{g\in\mathcal{T}_{\beta_n}\mathcal{V}_n}\left|\frac{1}{n}\sum_{i=1}^n |g(\mathbf{X}^{(i)}) - Y_{\beta_n}^{(i)}|^2 - \mathbb{E}[|g(\mathbf{X}) - Y_{\beta_n}|^2]\right|\right] + 2\mathbb{E}\left[\inf_{g\in\mathcal{T}_{\beta_n}\mathcal{V}_n}\int_{[0,1]^p}|g(\mathbf{x}) - \mathbb{E}(Y|\mathbf{X} = \mathbf{x})|^2 d\mathbf{x}\right].$$

Using Propositions C.3 and C.4 , we deduce that

$$\mathbb{E}\left[\mathbb{E}[|\widetilde{T}_s^{(n)}(\mathbf{X}) - Y|^2|\mathcal{D}_n]^{1/2} - \mathbb{E}[|\mathbb{E}(Y|\mathbf{X}) - Y|^2]^{1/2}\right] \leq 4a_n + 12b_n + 12\mathbb{E}[|Y - Y_{\beta_n}|^2] + \frac{12}{n}\sum_{i=1}^n \mathbb{E}[|Y^{(i)} - Y_{\beta_n}^{(i)}|^2].$$

### C.4.2 Consistency of the truncated estimator

**Theorem C.2.** *Set $M > 0$ and $\beta_n = \|\mathbb{E}(Y|\mathbf{X})\|_\infty + \tilde{\sigma}\sqrt{2}(\log n)^2$. Assume that a.s. $\mathbb{E}(Y|\mathbf{X}) \in H^s([0,1]^p)$ and that $(K_n)$ is a non decreasing sequence such that*

$$K_n \to \infty \ and \ \frac{K_n(\log n)^9}{n} \to 0. \tag{14}$$

*and that*

$$\max_{k=1,\cdots,K_n}\left[\mathrm{diam}(\mathcal{R}_k^{(n)} \cap [-M, M]^p)\right] \xrightarrow[n\to+\infty]{} 0.$$

*Then*

$$\lim_{n\to+\infty} \mathbb{E}[\tilde{T}_s^{(n)}(\mathbf{X}) - \mathbb{E}(Y|\mathbf{X})]^2 = 0.$$

**Proof:**
Let us first fix some non increasing sequence $(\eta_n)$ converging to 0. As we fixed $\beta_n$, we can choose $\lambda_n$ such that the following holds:

$$\beta_n > \frac{\lambda_n^{p/2}\|\mathbb{E}(Y|\mathbf{X})\|_{L^2([0,1]^p)}}{\inf_{|\boldsymbol{\omega}|\leq\lambda_n}|\mathcal{F}[\phi_{\boldsymbol{\sigma}}](\omega)|}.$$

Then, from Propositions C.3 and C.4 (Proposition 1 and Proposition 2 in the main paper), we deduce that

$$\mathbb{E}\left[\int_{[0,1]^p}|\widetilde{T}_s^{(n)}(\mathbf{x}) - \mathbb{E}(Y|\mathbf{X} = \mathbf{x})|^2 d\mathbb{P}(\mathbf{x})\right]$$

$$\leq \quad (4a_n + 12b_n + 12\mathbb{E}[|Y - Y_{\beta_n}|^2] + \frac{12}{n}\sum_{i=1}^n \mathbb{E}[|Y^{(i)} - Y_{\beta_n}^{(i)}|^2])^2$$

$$+2\mathbb{E}[|\mathbb{E}(Y|\mathbf{X}) - Y|^2]^{1/2} \cdot (4a_n + 12b_n(\beta_n) + 12\mathbb{E}[|Y - Y_{\beta_n}|^2] + \frac{12}{n}\sum_{i=1}^n \mathbb{E}[|Y^{(i)} - Y_{\beta_n}^{(i)}|^2])$$

where

$$a_n = C\|\mathbb{E}(Y|\mathbf{X})\|_{H^s([0,1]^p)}^2 \left(\frac{1}{(1+|\lambda_n|^2)^{s/2}} + \beta_n\left(\mathbb{E}\left[\max_{k\in\{1,...,K_n\}}\mathrm{diam}(\mathcal{R}_k^{(n)} \cap [-M, M]^p)\right] + M^{-r}\right)\right)^2,$$

$$b_n = \eta_n + 8\beta_n^2\exp\left[-\frac{n}{\beta_n^4}\left\{\frac{\eta_n^2}{2048} + \frac{\beta_n^4(K_n-1)\log(pn)}{n} + \frac{2K_n\beta_n^4}{n}\log\left(\frac{C'\beta_n^2}{\eta_n}\right)\right\}\right].$$

One assumes that $\eta_n = 1/\sqrt{\log n}$. The definition of $\beta_n$ and Condition (14) both imply that

$$\beta_n \to \infty \text{ and } \frac{K_n \beta_n^4 \log(\beta_n) \log n}{n} \to 0.$$

We then deduce that $b_n \to 0$. Letting first $M$ tend to infinity and thereafter choosing $n$ so that $\max_{k \in \{1,\ldots,K_n\}} \mathrm{diam}(\mathcal{R}_k^{(n)} \cap [-M, M]^p)$ is sufficiently small implies also that $a_n \to 0$ as $n \to \infty$. Then,

$$\lim_{n \to +\infty} \mathbb{E}[\tilde{T}_s^{(n)}(\mathbf{X}) - \mathbb{E}(Y|\mathbf{X})]^2 = 0.$$

### C.4.3 Extension to the untruncated estimator

Now we consider the untruncated estimator $\hat{T}_s^{(n)}$. From arguments similar to [Scornet et al., 2015] we are able to extend the consistency to the untruncated estimator:

$$\mathbb{E}[\hat{T}_s^{(n)}(\mathbf{X}) - \mathbb{E}(Y|\mathbf{X})]^2 \leq \mathbb{E}[\hat{T}_s^{(n)}(\mathbf{X}) - \tilde{T}_s^{(n)}(\mathbf{X})]^2 + \mathbb{E}[\tilde{T}_s^{(n)}(\mathbf{X}) - \mathbb{E}(Y|\mathbf{X})]^2.$$

We focus on the difference between truncated and untruncated estimors. As $\beta_n = \|\mathbb{E}(Y|\mathbf{X})\|_\infty + \tilde{\sigma}\sqrt{2}(\log n)^2$,

$$\mathbb{E}[\hat{T}_s^{(n)}(\mathbf{X}) - \tilde{T}_s^{(n)}(\mathbf{X})]^2 \leq \mathbb{E}\left([\hat{T}_s^{(n)}(\mathbf{X}) - \tilde{T}_s^{(n)}(\mathbf{X})]^2 \mathbb{1}_{\tilde{T}_s^{(n)}(\mathbf{X}) \geq \beta_n}\right)$$

$$\leq \mathbb{E}\left(\left[2\|\mathbb{E}(Y|\mathbf{X})\|_\infty^2 + 2\max_i |\varepsilon_i|^2\right] \mathbb{1}_{\max_i \varepsilon_i \geq \tilde{\sigma}\sqrt{2}(\log n)^2}\right)$$

because $|\tilde{T}_s^{(n)}(\mathbf{X})| \leq \|\mathbb{E}(Y|\mathbf{X})\|_\infty + \tilde{\sigma}\sqrt{2}(\log n)^2$. Then, following [Scornet et al., 2015] again, we get

$$\mathbb{E}[\hat{T}_s^{(n)}(\mathbf{X}) - \tilde{T}_s^{(n)}(\mathbf{X})]^2 \leq \frac{2\mathbb{E}(\|\mathbb{E}(Y|\mathbf{X})\|_\infty^2)n^{1-\log n}}{2\sqrt{\pi}(\log n)^2} + 2\left(3n\tilde{\sigma}^4 \frac{n^{1-\log n}}{2\sqrt{\pi}(\log n)^2}\right)^{1/2}.$$

Then,

$$\lim_{n \to +\infty} \mathbb{E}[\hat{T}_s^{(n)}(\mathbf{X}) - \tilde{T}_s^{(n)}(\mathbf{X})]^2 = 0,$$

which concludes the proof of Theorem 1 of the main paper.

## Footnotes

[1]available at `archive.ics.uci.edu/ml/datasets.php`

[2]available at `www.gagolewski.com/resources/data/ordinal-regression`

[3]available at `http://lib.stat.cmu.edu/datasets/boston`

[4]available at `https://www4.stat.ncsu.edu/ boos/var.select/diabetes.tab.txt`

[5]available at `https://www.rdocumentation.org/packages/missMDA/versions/1.14/topics/ozone`

[6]available at `https://www.csie.ntu.edu.tw/ cjlin/libsvmtools/datasets/regression.html`

[7]available at `http://dash.ipv6.enstb.fr/dataset/transcoding/`

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

[Supplementary Material 2]

scikit

learn 0.2

Corporate Identity Manual

# typography

scikit

*learn 0.2*

**corporate color**

#29ABE2 (CYAN)

#9B4600 (BROWN)

#F7931E (ORANGE)

scikit
learn 0.2

other versions