[Reviews · NeurIPS 2020]

Review 1

Summary and Contributions: A variation on regression trees is proposed that allows for fitting of smooth functions, rather than piecewise constant functions. The paper proposes a principled approach wherein the basis function at a leaf of the tree consists of the convolution of a probability density function and the indicator function of the region. This is in contrast to the standard regression tree, where the basis function is the indicator itself. Given these basis functions, the coefficients can be calculated with OLS regression of the outcomes on the basis expansion of the covariates. The tree structure itself is determined with a partitioning scheme similar to the CART procedure, except that split points are assessed based on their reduction of the global loss. In principle this requires performing OLS on each possible split point; hence the authors recommend limiting the set of candidate variables to a small number, choosing the best ones based on the standard CART impurity reduction. The method is proved to be consistent under a smoothness assumption on the conditional expectation function, requiring that the number of regions goes to infinity at the proper rate, and that the diameter of the regions goes to zero. The authors use their PR trees as the base learner in both random forests and a boosted ensemble. They compare the base PR and the ensemble variants to standard regression trees, random forests, and GBRTs. They also compare to other smoothed regression tree variants. The performance of the PR tree-based methods is very favorable.

Strengths: The authors have made a very convincing case for the effectiveness of this method. They use a variety of experimental datasets, on which their method is optimal or near optimal in both the base and ensemble variants. The consistency result provides a strong theoretical grounding to the method.

Weaknesses: The main drawback of the method seems to be the added computational complexity of the more complex splitting procedure (OLS regression at every candidate point). The authors mitigate this with their Top V procedure, showing that it does not sacrifice much predictive performance. Additionally, actually computing the fitted function at a single point requires evaluating a number of probability distribution functions (twice for each dimension of the input, for each leaf node). While this latter, test-time complexity is not discussed in the paper, it seems it could be substantially higher than the standard regression tree. The authors successfully performed experiments on sizable datasets (up to 70000 observations, and 9000 covariates), so it is clear that the method is not intractable. However, I would suggest the authors provide a plot of runtime (both training and test) for their method on the simulation datasets, both for base and ensemble variants, so the user can better understand the computational tradeoffs.

Correctness: The paper is technically correct as far as I can tell. I was not able to verify all the details of the consistency proof, due to time limitations for this review and the significant technical complexity of the proof.

Clarity: The method and experimental results are presented very clearly. I especially appreciate the significance analyses performed on the experimental results. In reading the theoretical results, I was a bit confused by their setup. Is the function space V_n a fixed or random function space (i.e. are we to consider the regions R_k fixed or random)? If fixed, I do not understand why there is an expectation in the definition of a_n in proposition 1. If random, then the theorem 1 seems to make assumptions on the random function space, which ought to be proven. If random, it is not discussed whether the regions are estimated from the same data as is used for the regression (honest trees), or whether the proof allows for the same data to be used for both.

Relation to Prior Work: The authors compare to the relevant prior work in smooth regression trees, as far as I know. The authors might consider comparing to methods that smooth out trees with a linear regression function, such as m5 and local linear forests.

Reproducibility: Yes

Additional Feedback: *** Comments after feedback received *** I appreciate the authors having responded to my concerns. Regarding computational complexity of least squares: Whether using (pseudo) matrix inversion, or Cholesky decomposition, I would assert there is an O(K^3) component. However, I would agree that the authors can consider O(K^2 n) to be the dominant term because this is a fixed-K analysis, so the authors are correct that O(K^3) can be ignored. Regarding the assumptions for Prop 1: thank you for the clarification that the regions R_k are fixed. I would suggest the authors address this more clearly in the statements of their theorems, as previous tree / random forest theory is very careful about assumptions like 'honesty' or data splitting. A major contribution of Scornet, Biau, Vert [22] is in the analysis that allows for data-driven split-point selection. However, this work requires that we consider split-point selection to be conditioned on. *** End of edit *** I was a bit confused by the "computation considerations" discussion. The authors state that (P^TP)^(-1) P^T requires O(K^2 n) operations. I don't understand how this is achieved, considering the standard matrix inversion would require O(K^3). Additionally, as mentioned, a discussion of test-time complexity would be useful. I would personally suggest a different title for the paper. The phrase “probabilistic regression trees” to me implies that there is a probabilistic model for the trees themselves (as in BART). However, this model is still solidly in the frequentist domain, but uses probability densities as convenient basis functions.


Review 2

Summary and Contributions: I acknowledge that I've read the rebuttal and I appreciate the effort that authors put in it. Experimental benchmarks and baselines are one of the main issues here. In order to make the method practical more evaluations must be performed. I'll keep my original score. ------ This paper proposes a method for learning decision trees for regression task by combining the idea of soft trees and greedy tree inducing. The resulting trees produce smooth predictions based on probability functions which is one of the main motivation behind this work.

Strengths: In general, I enjoyed reading this paper. The proposed method capitalizes on employing soft (a.k.a probabilistic) decision trees. However, PDE of each leaf (or region) is computed using non-parametric approach. I liked the idea of directly assigning a probability for the point reaching a particular leaf (region) rather than computing it along the path at each internal node by applying sigmoid like function which is commonly done is soft trees. Moreover, authors did a great job at investigating various aspects of their method. Specifically: 1. Theoretical: authors prove that the proposed trees are consistent meaning expected error approaches 0 as the number of points tends to infinity. But believe that this is expected behavior for such type of models (i.e. which use some form of averaging). For instance, RFs known to be consistent [1]. This work also uses some form of averaging since they employ probabilistic trees and thus this doesn't surprise me. However, it is nice to have it formally derived here. [1] Gerard Biau, Luc Devroye, and Gabor Lugosi. Consistency of random forests and other averaging classifiers. 2. Experimental evaluation: the method is tested on several regression benchmarks and shown to be performing well. However, baseline methods seems to be either outdated or too simple. For example, more advanced methods have been proposed recently on learning soft trees (see below [2-4]). Consider citing them and using some of them as a benchmark. My main concern is that the traditional trees already perform better/comparable against more sophisticated benchmarks. [2] Tanno et al. Adaptive neural trees. ICML'19 [3] Kontschieder et al. Deep neural decision forests. ICCV’15 [4] Forsst & Hinton. Distilling a Neural Network Into a Soft Decision Tree. 2017 3. Complexity: authors provide runtime analysis. Moreover, actual runtimes on several datasets can be found in suppl. mat. However, I'd be interested to see scalability of this model. The given datasets seems to be small (either # of instances or feature dimensions) for the current standards. Also, what about the model size (# of params) and inference time? In terms of model size, I think it it should be similar since the trees are axis aligned as in traditional trees. But in terms of inference time, it is clear that the traditional trees have advantageous of conditional computation, i.e. each instance follows a single root-to-leaf path which makes inference time logarithmic. Whereas, probabilistic trees were used in this method. On top of that, probability density function should be evaluated for each region. I think that the authors should mention and discuss such issues. Moreover, experimental session should contain such comparison/analysis. 4. Authors extend their analysis for an ensemble of such trees. ------ Finally, I would like to add here my thoughts on smoothness of the predictions. As far as I understand, it is one of the main motivations of this work. I tend to agree that the traditional CART style (a single) tree with constant leaves have a problem of non-smoothness since it separates the output space into the piece-wise constant regions. Moreover, the possible number of outputs is finite. However, I'm convinced that a single tree is rarely used in practice and they are mostly used as base learners in various ensembles (like RF). And this problem of non-smoothness kind of disappears when we use Random Forests. Yes, the possible number of outputs is still finite but it is exponential to the number of trees used in an ensemble. On top of that, we apply averaging. Therefore, RFs behave like a "pseudo-smooth" functions in that sense. The same is true for boosting. In short, I don't see smoothness of the output as a huge problem here.

Weaknesses: I combined strengths/weaknesses under the same section above. Sorry if it confuses.

Correctness: I don't see any problems in methodology and in theoretical part. See above for empirical evaluations.

Clarity: The paper is well written and easy to follow.

Relation to Prior Work: Yes.

Reproducibility: Yes

Additional Feedback: I would add somewhere a pseudocode.


Review 3

Summary and Contributions: I thank the authors for their rebuttal with convincing responses to most of the negative points i pointed out earlier. I am improving my evaluation score by one point to "marginally above acceptance threshold". - This paper addresses the problem that standard regression trees are limited to estimating piecewise constant linking functions, which might be a problem for estimating smooth functions linking input to output. - While in standard regression trees, a sample is assigned to only a single leaf, this paper proposes a method denoted as Probabilistic Regression trees (short: PR trees) to obtain soft leaf assignments, such that a sample can be assigned to multiple leaves. - The indicator function of the standard (hard) assignment is replaced by a probability distribution that indicates how close a sample is to all leaves. The authors propose to parametrize the pdf of each input feature belonging to a respective leaf by a location and scale parameter (e.g. a Gaussian). To obtain the joint distribution of the input vector x, the individual pdfs are multiplied. - The method is illustrated on a synthetic 1D example and tested on 13 regression benchmarks. - It is also demonstrated that the proposed method can be plugged into standard ensemble methods (Random forests and GBT)

Strengths: - Evaluated on several regression benchmarks - Comparison to existing decision tree methods - Show that the proposed method can be plugged into popular ensemble methods - Good overview of related work.

Weaknesses: - It seems there are underlying assumptions that are not clearly explained regarding equation 3 in Section 3: Input features x_j need to be independent, Otherwise the joint distribution of the input vector x belonging to regions R_k cannot be obtained by multiplying the individual pdfs of the input features x_j belonging to region R_k (note: region R_k equals a leaf k). If the same pdf is used for all x_j, then the assumption is that the input features x_j are iid (this is the case here) - Explain how the location parameters u_j is obtained. Is it the center of a region? - If enough data is available, a standard regression tree with enough leaves would also approximate a smooth link function. Maybe this approach would be advantageous in the small data regime? - The proposed method seems closer to “Uncertain Decision Trees (UDT)” than to “Soft trees” or “Smooth transition trees (STR)”. While Soft trees and STR have a soft branching mechanism based on a sigmoid at each node, the proposed PR trees have a soft assignment only at the leaves (last nodes). Uncertain Decision Trees propose also a soft assignment at the leaves, but use a different parametrization of the pdf (estimate the probability mass for discrete bins). Hence, it might be meaningful to compare the performance of PR trees to UDT. - The discussion of the empirical results is not convincing and contains some confusing statements regarding potential overfitting, please improve. - I think this work would get more attention in a journal format, which would give more room for detailed explanations.

Correctness: The experimental setup and evaluation procedure is fine. L. 228 – 232: This statement is confusing: When the number of training samples is large enough, then overfitting due to high model complexity should not be the problem. Overfitting could be a possible explanation for the performance drop of high capacity methods on the small datasets.

Clarity: The paper is written in a good language. However, some explanations e.g. for Equation 3 are too short. Please clarify phi and the parametrization (location and scale). Also clarify how the joint distribution of the input vector x is computed. In Eq. 3: is “1/sigma” not inside phi? If we chose e.g. a Normal distribution for phi, then 1/sigma is inside phi. It might be worth to rewrite Eq. 3. E.g. stating first that phi(x_j; u_j, sigma_j) is the probability of x_j being inside region R_k. then by assuming A we obtain the joint distribution D.

Relation to Prior Work: Related work is well explained, such that the reader gets a good overview. However, the authors claim their proposed approach is similar to Soft tress and STR trees, while I think it is actually closer to UDT. The advantage among existing work is not that clearly described. L.50 – 53: I do not understand this explanation of the drawbacks of previous work. In classification for example, the membership score (e.g. probability belonging to class k) is usually available by using a one-hot encoding of the true class label. L. 66: “The main difference between these approaches (Soft trees and STR trees) and PR trees lie in the way the assignment of examples to leaves is done, in number of parameters and in their theoretical guarantees.” L. 69: “…PR trees rely on less parameters which make them a priori more robust to overfitting.” Please clarify this statement. Why do Soft trees and STR trees have more parameters than PR trees? Regular regression trees have even less parameters than PR trees. Are PR trees more prone to overfitting than regular regression trees?

Reproducibility: Yes

Additional Feedback: - Mention that regions and leaves are the same - L. 5: change “how far”  “how close” - L. 29: “…preserving the interpretability…” Please elaborate on this. Interpretability due to the hierarchical decisions (like standard decision trees)? - L. 85: change “sum-of-trees”  ”ensembles of trees” - L. 86: clarify that a region R_k corresponds to a leaf - L. 87: Gamma is denoted “weight”, which I find a bit confusing. The leaf or region estimate would be more intuitive. - L. 91: “…piece-wise functions, may fail to accommodate the smoothness of the link function.“  maybe explain and clarify the setting when that is a problem, e.g. small training dataset in a high-dimensional feature space. Otherwise, with enough training data and enough leaves, even a regular regression tree can approximate a smooth link function. - L. 242: This Figure has no caption - Figure 3: Describe the meaning of the vertical dotted lines.


Review 4

Summary and Contributions: * a new probabilistic version of decision trees * theoretical results including a consistency proof * benchmark experiments showing good results

Strengths: + a neat idea in an extensively studied topic + solid looking theory + good empirical results

Weaknesses: - not exactly clear how much better the proposed method is than random forest - computational complexity compared to random forests wasn't (apparently) discussed – this is a major question (if RF is significantly more efficient, the argument in favour of the proposed method is much weaker than it may seem)

Correctness: yes, as far as I can tell

Clarity: yes, well enough

Relation to Prior Work: seems to be. the authors didn't discuss the relationship to BART, but I think that's reasonable since they do discuss relationship to random forests which are computationally more efficient and more commonly used (as far as I can tell)

Reproducibility: Yes

Additional Feedback: ***remarks added after rebuttal*** Right, now I get why the Gaussian kernel with \sigma->0 reproduces regular decision trees – I was confusing the u and the x in the integral defining \Phi, so that thought that Gaussian density isn't properly normalized, but it is.) I'm still a bit concerned by the computation vs accuracy tradeoff compared to random forests. The authors provide computation time figures for simple decision trees where the proposed method is about 100-1000x slower. Random forests (in the regression case with default p/3 active features) would probably be somewhat faster assuming the number of trees is of the order 100. But in any case, the comparison is not totally lopsided in favor of RFs, so I'm quite satisfied with this aspect. I was assuming this not to be a serious issue, so I already chose my overall rating to be positive, and I will keep it unchanged. ***end of post-rebuttal comments; note: the following comments haven't been revised after the rebuttal*** Something I was left wondering about in the construction, Eq. (3), is whether this can reproduce standard regression trees. In other words, is there a density function \phi that implies \Phi(x) = 1_{x\in\mathcal{R}_k}, where 1_{} is the indicator function; see line 96. Overall, I'm wondering if there is a connection to kernel regression, radial basis networks, and mixture models, since the proposed method produces predictions by weighting components based on, essentially, a kernel density function \phi.

[Author Response · NeurIPS 2020]

**General response.** First of all, we thank the reviewers for their helpful comments and remarks. We provide first general
comments prior to address additional points raised by the reviewers.

We want to stress that we have proposed a generalization of regression trees that (1) adapt to the smoothness of the
prediction function relating input and output variables while (2) preserving the interpretability of the prediction and (3)
being robust to noise. The three points, smoothness, interpretability and robustness to noise, are all important and have
been illustrated empirically. There is however *no free lunch*, and these additional properties come with a computational
cost, as described for training in Appendix A.2 (note that, as mentioned in the main paper line 193, we make use of the
Moore-Penrose pseudo-inverse which explains why the complexity is only quadratic in K). Applying PR trees is also
more costly than applying standard decision trees as the function $\Psi$ (Eq. 2) needs to be evaluated on all regions. We
provide below the prediction time, in seconds, on some datasets (we'll include these results in Appendix A.2).

| Dataset | PR Tree | Std Tree | # Observations |
|---------|---------|----------|----------------|
| BD | 0.3 | 1.00E-04 | 146 |
| BO | 0.24 | 1.00E-04 | 101 |
| DI | 0.2 | 9.00E-05 | 88 |
| RI | 0.04 | 2.00E-04 | 14 |

We also want to emphasize that the theoretical framework we propose does not assume that the $x_j, 1 \leq j \leq p$, are
independent. The notation $\phi\left( ()_{1 \leq j \leq p} \right)$ in Eq. 3 means that $\phi$ is a multivariate function of the $p$ variables $\frac{u_j - x_j}{\sigma_j}$. For
convenience, we have used functions $\phi$ that lead to standard cdfs for $\Psi$ in our experiments, dropping the dependencies
between $x_j$. Other choices could be made, in particular when dependencies between $x_j$ are known. In any case, $u_j$
cannot be interpreted as a location parameter or as a center of a region as it is the variable that is integrated out.

**Reviewer 1.** Ooops, you are right: The expectation in the expression of $a_n$ in Proposition 1 should be removed (this
proposition directly derives from Proposition C2 in Appendix C2.2, with no expectation; the expectation should also be
removed from Proposition C3 in Appendix 2.4). The regions $\mathcal{R}_k^{(n)}$ are fixed for a given $n$.

**Reviewer 2.** An important difference wrt to the work by Gérard Biau, Luc Devroye and Gabor Lugosi ([1]) is that we
are not averaging over independent classifiers as regions are dependent on each other. Our consistency proof radically
differs from theirs because of this difference.

Adaptative Neural Trees ([2]) and Deep Neural Decision Forests ([3]) are both built from decision trees. These models
are very close to soft trees, to which we compare ourselves. In each case however, the models are enhanced with
a neural network representation and suffer from a lack of interpretability (one can even argue that these models are
not tree models *per se*). The paper of Forsst & Hinton ([4]) considers a specific variant of the soft tree model, with
knowledge distillation. Distilling knowledge into our trees is clearly an interesting research direction that we plan to
investigate.

Because of their interpretability, decision trees seem to be still heavily used in the industry, as mentioned in the 2019
Kaggle survey (https://www.kaggle.com/kaggle-survey-2019). This said, Random Forests aim at reducing the variance
(and this comes at the expense of a small increase in the bias) whereas our adaptation to smoothness aims at reducing
the bias. Combining both, as in PR-RF, reduces both bias and variance and leads to a method which significantly
outperforms RF (Table 5, Appendix A.4).

**Reviewer 3.** It is true that a standard regression tree with enough leaves can also approximate a smooth link function.
However, to obtain such a tree, one needs large samples, which are unfortunately not available in practice (as examplified,
*e.g.*, by the difference between standard and PR trees in our experiments).

Uncertain decision trees were designed to deal with uncertainty in the input variables and rely on a set of given pdfs
modeling the uncertainty on each attribute value for this particular example. This contrats with our approach that aims
at adapting to the smoothness of the prediction function. In particular, the intervals $[a_{i,j}, b_{i,j}]$ (reference [24] of our
paper) defining the support of the pdfs are given in uncertain decision trees whereas they are learned in our case.

Our discussion on overfitting simply amounts to saying that the more complex a model is, the more likely it is to overfit
(in practice, the amount of samples available is usually not large enough to avoid that). We'll modify lines 228-230 as
we agree that they may be confusing. The additional complexity of PR trees compared to standard trees is not important
and has not led to overfitting in our experiments.

There is a typo in line 194 as it is $\Psi$ (and not $\phi$) that corresponds to the cdf of a normal distribution (multivariate normal
distribution with diagonal covariance matrix equal to $\boldsymbol{\sigma}$).

**Reviewer 4.** One can obtain standard regression trees from Eqs 2 and 3 by setting $\phi$ to $(2\pi)^{-\frac{p}{2}} \prod_{j=1}^{p} \exp(-\frac{(u_j - x_j)^2}{2\sigma_j^2})$,
with $\sigma_j \to 0$ for all $j$. In that case, the distribution of $\mathbf{x}$ over regions is concentrated on one region.

[Meta-Review · NeurIPS 2020]

The reviewers uniformly felt that this is an interesting paper, and a good contribution to the community.